# Discriminating viscous creep features (rock glaciers) in mountain permafrost from debris-covered glaciers – a commented test at the Gruben and Yerba Loca sites, Swiss Alps and Chilean Andes

Wilfried Haeberli[1], Lukas U. Arenson[2], Julie Wee[3], Christian Hauck[3], and Nico Mölg[4]

[1]Geography Department, University of Zurich, 8057 Zurich, Switzerland
[2]BGC Engineering, Suite 500 – 980 Howe Street, Vancouver, BC V6Z 0C8, Canada
[3]Department of Geosciences, University of Fribourg, 1700 Fribourg, Switzerland
[4]enveo, Fürstenweg 176, 6020 Innsbruck, Austria

*Correspondence to*: wilfried.haeberli@geo.uzh.ch

**Abstract.** Viscous flow features in perennially frozen talus/debris called rock glaciers are being systematically inventoried as part of global climate-related monitoring of mountain permafrost. In order to avoid duplication and confusion, guidelines were developed by the International Permafrost Association for discriminating between the permafrost-related landform "rock

glacier" and the glacier-related landform "debris-covered glacier". In two regions covered by detailed field measurements, the corresponding data- and physics-based concepts are tested and shown to be adequate. Key physical aspects, which cause the striking morphological and dynamic difference between the two phenomena/landforms concern:

- tight mechanical coupling of the surface material to the frozen rock-ice mixture in the case of rock glaciers as contrasting with essential non-coupling of debris to glaciers they cover;

- talus-type advancing fronts of rock glaciers exposing fresh debris material from inside the moving frozen bodies as opposed to massive surface ice exposed by increasingly rare advancing fronts of debris-covered glaciers; and

- increasing creep rates and continued advance of rock glaciers as convex landforms with structured surfaces versus predominant slowing down and disintegration of debris-covered glaciers as often concave landforms with primarily chaotic surface structure.

Where debris-covered surface ice is, or has recently been, in contact with thermally-controlled subsurface ice in permafrost, complex conditions and interactions can develop morphologies beyond simple "either-or"-type landform classification. In such cases, remains of buried surface ice mostly tend to be smaller than the lower size limit of "glaciers" as applied in glacier inventories, and to be far thinner than the permafrost in which they are embedded.

# 1 Introduction

Application of modern technologies enables climate-related science and long-term monitoring of mountain permafrost on Earth to evolve in a rapid, fascinating and important way. Absolute age determination using radiocarbon, luminescence or exposure dating (e.g., Haeberli et al., 2003; Fuchs et al., 2013; Krainer et al., 2015; Nesje et al., 2021; Amschwand et al., 2021) document the multi-millennial, typically Holocene time scale related to the often-spectacular landforms usually called "rock glaciers", which reflect cumulative deformation through slow viscous creep of perennially frozen talus/debris rich in ice

(Wahrhaftig and Cox, 1959; Haeberli, 1985; Barsch, 1996; Haeberli et al., 2006, Berthling, 2011; Kääb, 2013). Sophisticated borehole observations and laboratory experiments together with results from high-precision geodetic-photogrammetric and interferometric measurements (e.g., Arenson et al., 2002; Roer et al., 2008; Springman et al., 2012; Cicoira et al., 2021; Kääb et al., 2021; Kaufmann et al., 2021; Cusicanqui et al., 2021; Noetzli et al., 2021; Thibert and Bodin, 2022; Fleischer et al., 2023) provide detailed and crucial quantitative information about the corresponding flow mechanisms and flow fields with

remarkable indications of recent warming-related flow acceleration.

Increasing creep rates of rock glacier permafrost can be attributed to permafrost warming down to tens of meters below surface as systematically measured in boreholes (Harris et al., 2009; Etzelmüller et al., 2020; Hoelzle et al., 2022). Warming-induced reduction in strength of subsurface frozen materials parallels increasing amounts of unfrozen water as reflected by repeat geophysics (Mollaret et al., 2019; Buckel et al., 2023), a phenomenon and long-term climate impact which

also affects the stability of perennially frozen rock walls (Davies et al., 2001; Gruber and Haeberli, 2007; Krautblatter et al., 2013; Mamot et al., 2021; Shugar et al., 2021). In striking contrast to enhanced creep rates in rock glacier permafrost, flow dynamics in lower parts of debris-covered glaciers are decreasing as a consequence of stronger thinning in their upper boundary zone with related reduction of driving stresses (Clayton, 1964; Anderson et al., 2021) and of continued warming-induced ice loss (Neckel et al., 2017).

In its responsibility for the Global Terrestrial Network for Permafrost (GTN-P) as part of the Global Climate Observing System (GCOS), the International Permafrost Association (IPA) undertakes focused efforts via an Action Group (Rock Glacier Inventories and Kinematics; RGIK, 2023) to internationally coordinate the development and compilation of rock glacier inventories and the long-term monitoring of rock glacier kinematics (RGIK, 2023). This action group consisting of numerous experts from around the world develops, through consensus, technical guidelines including recommendations on

how to discriminate the landform "rock glacier" as expression of viscous creep in perennially frozen subsurface materials (Haeberli et al., 2006; Cicoira et al., 2021; Arenson et al., 2021) from debris-covered glaciers with their surface ice monitored as part of the Global Terrestrial Network for Glaciers (GTN-G). An objective way of differentiating corresponding landforms and kinematics is essential in creating clarity when utilizing such landforms to assess where and how climate change impacts our planet, specifically the cryosphere, or when used in a regulatory/legal context, for example in view of hydrological

significance; or generally, to avoid confusion and duplication. The present note attempts at testing the proposed technical recommendations/guidelines and at commenting on them using two practical examples: i) the Gruben site in the Swiss Alps

(Figure 1) and ii) the Yerba Loca Valley in the Chilean Andes (Figure 5). At the Gruben site, detailed and comprehensive field investigations of permafrost, glaciers and complex contacts between the two have been carried out for more than half a century in connection with hazardous lakes and flood-protection work (see for details Haeberli et al., 2001; Gärtner-Roer et al., 2022; Wee et al., 2022). While less information exists for the Yerba Loca Valley, it is a valuable site for which some data is available, and it helps demonstrating that similar conditions exist in different mountain ranges.

## 2. Terminology, characteristics and guidelines:

The RGIK document provides rich, detailed and comprehensive explanations based on consensus of experts from around the world. As a short summary of this important source, the following terms and guidelines are adopted here and completed with brief notes on principles of surface and subsurface ice with their geophysical characteristics.

*Terms:*

Landforms: Rock glaciers are debris landforms generated by the former or current creep of frozen ground (permafrost; RGIK 2023; cf., Haeberli et al., 2006; Berthling, 2011), detectable in the landscape with the following morphologies: front, lateral margins and optionally ridge-and-furrow surface topography. In coherence with global glacier inventory standards, the minimum rock glacier size to be included into a global compilation should be 0.01 km$^2$. Rock glaciers should not be confused with debris-covered glaciers, which are glaciers partially or completely covered by supraglacial debris. The discussion in our invited perspective focusses (a) on patterns of ridges and furrows as expressions of cohesive flow, and (b) on frontal characteristics rather than lateral margins.

*Surface and subsurface ice:*

The term "glacier" is explicitly defined as "on the land surface" (Cogley et al., 2011), i.e., as surface ice. Therefore, debris-covered glaciers as contained in glacier inventories are, by definition, surface ice. Characteristic forms of surface ice, here also in the sense of debris-covered surface ice, are differentiated between glaciers, glacierets, perennial ice patches and dead ice bodies, because the term "glacier" is not appropriate for most of the commonly small landforms in question for reasons of size (area, elevation range) and dynamics. The definition of the term "permafrost" explicitly relates to thermal conditions of "ground (soil or rock and included ice or organic material)" (International Permafrost Association [IPA], 2023), i.e., of subsurface materials. Ice contained in permafrost is, therefore, by definition subsurface ice or ground ice, independent of its spatial extent. The term permafrost is thereby independent of material characteristics, i.e., it is not "a type of ground" but a definition of its thermal state.

Confusion sometimes arises from the use of the term "glacier" in the misleading but historically established and today generally accepted term "rock glacier" as applied to a landform created by subsurface ice under thermal conditions of permafrost. Such confusion can be avoided by accompanying the term "rock glacier" with process- and material-related expressions like "viscous creep features in mountain permafrost", as done in the title of the present contribution.


*Geophysical characteristics:*

Geophysical characteristics of perennially frozen subsurface materials and of massive (debris-covered) surface ice show marked differences which can be summarized as follows:

Thermal conditions of rock-glacier permafrost are measured in boreholes (Noetzli et al., 2021), by using miniature data loggers at shallow depth (PERMOS, 2023), or they can be approximated through applying numerical models based on climate data (e.g., Haq and Baral, 2019; Baral and Haq, 2020; Li et al., 2023) optimally in combination and, if possible, supported by geodetic measurements of flow characteristics to define activity levels (cf. Bertone et al. 2023). Results from extended time series document ongoing subsurface warming trends (Etzelmüller et al., 2020). Frozen conditions mostly reach down to depths 105 of tens of meters to more than 100 meters. Vertical temperature gradients and heat flow values at depth are strongly reduced due to historical and ongoing surface warming. Debris-covered surface ice can be temperate, polythermal or cold. Perennial ice patches from avalanche cones or glacierets are most common in connection with rock-glacier permafrost. Such small/thin and rather static surface ice bodies can be assumed to be predominantly cold, because their ice cannot warm up above 0°C during the warm season but cool down far below 0°C during the cold part of the year.


In view of geophysical soundings (e.g., Haeberli and Vonder Mühll, 1996; Hausmann et al., 2007; Hauck and Kneisel, 2008; Merz et al., 2015; Pavoni et al. 2021; Halla et al., 2021; de Pasquale et al., 2022), ice-sediment mixtures in rock-glacier permafrost tend to produce

• strong scatter causing reduced transparency for electromagnetic waves, while homogenous/massive surface ice – especially if cold – is highly transparent for radio-echo soundings;
• heterogenous patterns of seismic P-wave velocities with characteristic values varying mostly between about 2,500 and 4,500 m/s, while homogenous/massive surface ice exhibits more uniform values close to 3600 m/s:
• heterogenous patterns of electrical D.C. resistivities with characteristic values ranging from about 10 kΩm to near 1 120 MΩm, Herring et al., 2023), in contrast to massive surface ice with characteristic values of 1 to 10 MΩm for small ice patches and glacierets primarily consisting of superimposed ice, and of >100 MΩm for glacier ice from warm/wet firn metamorphosis with efficient ion evacuation by percolating meltwater.

*Guidelines for landform interpretation:*

In addition to detailed qualitative explanations, RGIK (2023) provides the following check-list table for discriminating rock glaciers from debris-covered glaciers or other forms of surface ice:

| Geomorphological/ Kinematic feature | Rock glacier | Debris-covered glacier |
|---|---|---|
| Transverse ridges and furrows | Frequent | Non-frequent |
| Talus-like front | Frequent | Non-frequent |
| Crevasses with exposed ice | Non-frequent | Frequent |
| Abundant thermokarst | Non-frequent | Frequent |
| Abundant supraglacial lakes | Non-frequent | Frequent |
| Ice cliffs | Non-frequent | Frequent |
| Supraglacial streams/channels | Non-frequent | Frequent |
| Subsidence rate | $\sim$cm/y$^{-1}$ | $\sim$m/y$^{-1}$ |
| Flow field coherence | Good (unless too fast) | Reduced, due to differential melt |

**Table 1: Indicative features to distinguish between rock glaciers and debris-covered glaciers (RGIK, 2023).**

## 3 Rock glacier and cold debris-covered glacier at the Gruben site

The continuously advancing *Gruben rock glacier* (Figure 1) with its strikingly coherent flow field is a convex landform exhibiting a regular surface pattern of ridges and furrows – here lengthwise following the predominantly extending flow –, and has over-steepened talus-like fronts and margins continuously exposing fresh material from the interior parts of the moving body. It neither exhibits ice cliffs, nor surface streams, visible crevasses exposing ice or thermokarst/surface lakes as it would be characteristic for debris-covered glaciers. Older, as well as recent geophysical soundings document permafrost tens of

meters deep (Figure 2; cf. supplement Figure SUP.-1) and still thermally active in that winter freezing reaches down through the active layer to the top of the permafrost (cf. ongoing GST-measurements reported by Gärtner-Roer et al., 2022). As a consequence of global warming, however, ground temperatures are currently in strong thermal imbalance, with only slightly negative near-surface temperatures and a near-zero vertical geothermal gradient/heat flow at depth as it appears to be characteristic for present-day mountain permafrost (Haeberli, 1985, cf. Etzelmüller et al., 2020). These geothermal conditions

have been confirmed for many permafrost sites in the Swiss Alps (Mollaret et al., 2019; Phillips et al., 2020; PERMOS, 2023) and are indicative of the degrading state of permafrost at this site causing characteristic rates of surface subsidence (thaw strain / settlement) in the range of centimeters per year. Due to the slow process of heat diffusion with latent heat exchange at depth,

complete thawing of the ice-rich rock glacier permafrost would most likely need centuries if not millennia (cf. early calculations reported by Haeberli, 1985).

In strong contrast, the continuously slowing-down and decaying debris-covered cold tongue of *Gruben glacier* (Figure 1) with its progressive loss of coherent flow is a concave landform with a chaotic rather than regular surface pattern, with diffuse and hardly recognizable frontal and lateral margins but with, in places, local ice cliffs, visible crevasses exposing ice and thermokarst/surface lakes. Detailed long-term measurements of horizontal and vertical surface displacements (Gärtner-Roer et al., 2022) for the past decades document characteristic subsidence rates in the range of tens of centimeters per year due

to ice melting. From earlier measured radio-echo soundings (Haeberli and Fisch, 1984) and borehole temperatures (Haeberli, 1976), the remaining ice underneath its debris cover can be estimated to be about -1 to -2°C cold and its present-day thickness to decrease from a maximum of about 20 m at the upper end of continuous debris cover to zero at the diffuse margins close to Lake 1 (Figure 2). Complete vanishing of the presently existing remains of the glacier tongue is most likely a matter of a few decades. Subglacial permafrost may presently penetrate into the thick sediment bed (Haeberli and Fisch, 1984) underneath the

rapidly thinning and vanishing cold glacier remains.

        The differences are obvious, and the morphological criteria (steep front and lateral margins, surface pattern with ridges and furrows for rock glaciers, etc.) proposed by the IPA action group are adequate: The subsurface ice contained in the striking viscous creep feature of *Gruben rock glacier* is a permafrost (periglacial) phenomenon and as such part of global permafrost monitoring. The increasingly indistinctive feature of the down-wasting debris-covered *Gruben glacier* tongue is a

(glacial) phenomenon of surface ice and as such part of global glacier monitoring. Comparison between the measured kinematics of the two features at this site documents strikingly different characteristics and dynamics of the two domains under conditions of global warming and highlights the importance of separating these two different features when developing inventories. The situation is clear and the differentiated treatment of the two phenomena within the framework of internationally coordinated global climate observation perfectly appropriate. Comments are, however, needed concerning two

special aspects:

- complex contact zones between debris-covered surface ice and thermally controlled subsurface ice; and
- the sometimes still maintained idea that covering of surface ice by debris alone can produce striking viscous flow features called "rock glaciers".

## 4 Complex zones with contacts between surface and subsurface ice

Where mean annual air temperatures are, or have until recently been, below zero centigrade, various types of surface ice - especially cold perennial snow fields or ice patches, glacierets and mostly small, cold to polythermal glaciers - can be in contact with thermally controlled, creeping ice-rich mountain permafrost, contributing debris and in cases remains of buried ice to deeply frozen materials. This had already been recognized by Wahrhaftig and Cox (1959), Fisch et al. (1978), Haeberli (1985) or Barsch (1996) and in the meantime became the subject of intense and detailed quantitative investigations by various authors

(cf., for instance, Kneisel and Kääb, 2006; Bosson and Lambiel, 2016; Monnier and Kinnard, 2017; Bolch et al., 2019; Falatkova et al., 2019; Kunz et al., 2021; Vivero et al., 2021; Wee and Delaloye, 2022; Wee et al., 2022). At the Gruben site, a former contact zone exists between the polythermal Gruben glacier during its maximum LIA/Holocene advances and the permafrost of the Gruben rock glacier (Figure 1; cf., Kääb et al., 1997; Gärtner-Roer et al., 2022). Contacts and interactions between glaciers and rock glaciers can give rise to a diverse range of landforms, exhibiting a wide spectrum of characteristics.

These landforms encompass structured surface morphology, such as glacitectonics or back-creep, which can either align with or oppose the original stress exerted by the advancing glacier. They can also encompass diffuse and often chaotic topography with thermokarst features in degrading or aggrading permafrost. Furthermore, the local surface coarseness can be smoothed due to the deposition of fine-grained sediments (Wee and Delaloye, 2022; Kneisel and Kääb, 2007).

Concerning the "transfer" of surface ice to creeping (rock glacier) permafrost, there is no simple or straightforward

general solution. The Gruben and Yerba Loca examples, however, provide some indications. As mentioned in the caption of Figure Sup.-1, the isolated bodies with resistivities in the low MΩm range, still existing today on top of near 0°C permafrost at Gruben in the former marginal zone of the LIA glacier are most probably dead ice from the small northern tributary underneath the Senggchuppa slope but could also be remains of a buried and frozen avalanche cone at the origin of the photogrammetrically defined flowlines. The earlier visible surface ice at Yerba Loca cannot be called "glaciers" for reasons

of size but are/have been perennial ice patches, mostly from avalanche cones. In both cases, Gruben as well as Yerba Loca, the buried ice bodies are more or less passively riding on top of thick perennially frozen sediments. The persistence of MΩm-buried ice in lower parts of rock glaciers are rare exceptions.

Such complex and highly variable landforms can mostly not be attributed in a straightforward "either-or" scheme to the terms "rock glacier" or "debris-covered glacier" but constitute what could better be called "complex contact zones of

(viscous creep in ice-rich) permafrost with remains of buried surface ice". Various forms of surface ice must thereby be treated in a differentiated way – not every piece of surface ice is a "glacier". A number of potentially interconnected effects and processes can take place as related to loading/unloading, reorientation of stress and flow fields, intermittent burial of surface ice in frozen debris with, in cases, thermokarst development due to its warming-induced vanishing, penetration of permafrost into previously unfrozen parts of now exposed glacier beds, etc. Understanding the full spatiotemporal complexity of relations

and interactions and heat fluxes between surface and ground ice in such areas by far exceeds possibilities of intuitive/simplistic landform interpretation from visual or remotely-sensed surface inspection alone. Combinations of sophisticated quantitative measurements are needed, especially geophysical soundings, geotechnical drillings with undisturbed core extraction and geodetic measurements, in order to define subsurface physical conditions (thermal, hydrological and mechanical; stress conditions), material properties and characteristics (especially ice content, unfrozen water, cohesion and internal friction) and

related physical processes (creep, shearing, cumulative long-term deformation, infiltration/advection, latent heat exchange) with their changes in time.

Neither the term "rock glacier" nor the term "debris-covered glacier" would be appropriate for such complex contact zones with their characteristically diffuse landforms. A practicable approach for internationally coordinated inventory work to

deal with such overlapping/combined contact zones still needs development. An important if not decisive aspect may thereby be that remains of surface ice buried in permafrost are – as the Gruben and Yerba examples document – mostly smaller than the lower size limit applied to the term "glacier" in glacier inventories and much thinner than related permafrost depths (cf. the Kintole conditions documented in Haeberli, 1985).

## 5 Debris-covered glaciers remain debris-covered glaciers

Only few authors continue postulating – in full contradiction to the rich measured evidence – that the burial of massive surface ice alone, i.e., in the absence or independently of long-term freezing at depth with its fundamental impact on subsurface material properties, can produce characteristic viscous creep features called rock glaciers (Anderson et al., 2018) or even that rock glaciers are nothing else than debris-covered LIA glaciers (Whalley, 2020; cf., the literature overview provided by Janke and Bolch, 2022, the community comments related to our contribution by Harrison (2023) and Whalley/Azizi (2023) with response from our side (Haeberli et al., 2023), and Gärtner-Roer et al. (2022) especially concerning the Gruben site). A never discussed implication of these beliefs from the side of intuitive landform interpretation is the rather astonishing, tacit assumption that debris-covered glaciers have two options of possible developments:

(a) they can remain *debris-covered glaciers* with predominantly chaotic surface structure, slowing down their flow, and decaying/disintegrating under conditions of global warming as increasingly concave landforms with diffuse margins; or

(b) they can become *rock glaciers* by maintaining or even accelerating their flow and advance under conditions of atmospheric temperature rise, adopting a coherent flow pattern of their surface debris and becoming convex landforms with strikingly "organized" surface structures and over-steepened talus-type margins/fronts continuously exposing fresh debris (instead of massive ice) from their inner parts.

Option (a) is documented in a rich, quantitative scientific literature (e.g., Iwata et al., 1980; Anderson and Anderson, 2018; Mölg et al., 2020; Shokory and Lane, 2023) and needs no further discussion. Version (b), on the other hand, is in full contradiction with the wealth of quantitative information available from decade-long sophisticated measurements all over the world. It is nevertheless useful to understand the reason why version (b) does not seem to occur in nature and what the physical causes are for the striking differences in appearance and dynamic evolution of rock glaciers and debris-covered glaciers. Could perhaps permafrost and cold ice conditions constitute a possibility for debris-covered glaciers to "learn" how to turn into something so strikingly different from what they are initially?

Debris covers on both, rock glaciers and glaciers, are usually predominantly coarse-grained. Frost-susceptible fine material tends to be washed out and/or to accumulate at the bottom of the debris cover/permafrost active layer. In addition, the source zones of the debris, which are typically steeper rock faces, favour the generation of coarse material. Excess ice formation inducing cohesion and reduced internal friction can therefore not take place throughout coarse-grained surface layers. As a consequence, material properties of the debris cover themselves can hardly make a decisive difference. Material properties,

however, exert strong influences where frost-susceptible fines freeze, build up excess ice, and remain frozen. The key for understanding the striking difference between the involved processes and structures, therefore, relates to the formation of the ice-debris combination and the corresponding mechanical and thermal coupling between the surface layer of debris and the moving mass underneath (Figure 3). While freezing processes induce the formation of ice inside the rock material and, hence, induce a tightly interconnected mixture of ice and rock particles, largely pure surface ice in debris-covered glaciers or ice patches forms first, and rock components covering it are added independently in a second step as a fundamentally different process and without any tight connection to the ice.

## 6 The key role of ice-debris coupling …

On *rock glaciers*, the rock components in the lower part of the "cover", which includes the thermally defined active layer, reach down beyond the permafrost table and are firmly frozen within the creeping mass. Unless the permafrost is in an advanced state of degradation, mean annual temperatures just below the permafrost table are substantially colder than zero centigrade, adding to the strength of this mixed rock-ice layer. The coherent movement pattern of the perennially frozen mass at depth with its excess ice in fine material, strong cohesion and reduced internal friction is thereby directly transmitted to the rock components of the active layer. Components of the upper active layer, which may intermittently reach positive temperatures during summertime, are firmly interlocked with the deeper, frozen-in components and, hence, directly coupled with the creep movement of the frozen body underneath. This effect is especially strong in zones of compressing flow, which generally occurs in lower and flatter reaches of rock glaciers, causing buckle folding and the creation of characteristic arcuate transverse ridges and furrows (Frehner et al., 2015). Where movement speeds increase along the flow direction, especially in steep root zones of rock glacier flow, extending flow with tensile stresses weakens the interlocking effect between rock components at the surface, thereby often making surface structures somewhat diffuse or producing longitudinal rather than transverse ridge-and-furrow structures. The decisive fact related to the striking viscous appearance of rock glaciers is the large-scale stress coupling which is transferred from the cohesive creep of the thick perennially frozen body at depth to the partially frozen-in and interlocked surface layer.

Components of the debris cover resting on the surface of *massive ice* have hardly any contact with the rare major rock components inside the ice underneath. They are, therefore, not directly coupled with the movement of the ice underneath but typically remain freely mobile at the ice-debris interface (Figure 3). Relative displacements of individual components at smallest spatial scales are easily possible and tend to follow spatially variable, local melt patterns of the underlying ice (cf., Alean et al., 2020), which are further impacted by water flow patterns that develop at the surface of the ice, and thermokarst (cryokarst) processes leading to the collapse of the overlying ice (and debris) layers (Thompson et al., 2016; Mölg et al., 2020). Rather than coherent flow patterns, the debris cover on glaciers or on other buried massive ice tends to develop chaotic surface structures following small-scale features of ground ice melt (cf. Kneib et al., 2023). The mostly chaotic surface structure of debris-covered glaciers is due to the non-transmission of large-scale stress-coupling from the massive ice body at depth to its

loose and unconstrained debris cover. This also applies to cold or polythermal ice conditions such as they exist at the Gruben site (cf. also Miles et al., 2018 for Khumbu glacier), because the thickness of the debris layer on glaciers (typically decimeters) is mostly much smaller than that of the thermally controlled active layer (typically meters) in the permafrost of rock glaciers. With other words: debris-covered glaciers have no choice – they remain debris-covered glaciers.

## 7 … as visible at advancing fronts …

A key element for understanding and recognizing the differences between creeping frozen debris and glaciers covered by debris is the morphology of over-steepened, advancing fronts (Figure 4).

Advancing fronts of creeping perennially frozen debris are presently widespread under conditions of atmospheric temperature rise and related permafrost warming/softening with resulting creep acceleration. Ongoing movements create their characteristic talus-like morphology with freshly exposed debris from inside the creeping mass in their over-steepened upper parts (cf., the detailed analyses by Wahrhaftig and Cox, 1959; Kääb and Reichmuth, 2005; Kummert et al., 2021). Rock glacier fronts show over-steepening in the upper section of the front because of the fines and the suction that is generated within the unsaturated sediments of the active layer. The slope of the lower part of the rock glacier front, as clearly visible on Gruben rock glacier (Figure 4), is typically at the angle of repose of the debris as these sediments accumulate in response to them being released from the upper part of the slope. Direct exposures of frozen materials are only observed in connection with extraordinary erosional processes such as detachment slides (Haeberli and Vonder Mühll, 1996; Arenson and Jakob, 2015) or extreme fluvial erosion (Elconin and LaChapelle, 1997).

Advancing debris-covered glaciers have become exceptional under conditions of atmospheric temperature rise and predominating glacier shrinkage. In such increasingly rare cases, massive ice of the flowing glacier is usually visible at near-vertical fronts where debris cannot accumulate (Figure 4). Massive ice is also often visible at terminal margins of shrinking debris-covered glaciers.

The advantages of interpreting over-steepened fronts of creeping frozen talus/debris can be illustrated with an example from the Andes of Chile (Yerba Loca Valley; cf. Marangunic et al., 2022). Numerous typical advancing fronts (yellow arrows in Figure 5) document active creep behavior of perennially frozen talus as confirmed by animations using recent aerial lidar scans (cf., supplement). Near points 4 and 5 in Figure 5, at an elevation of about 4150 m, a mean annual air temperature of -4.4°C was measured in 2014 and -2.5°C at 0.4 m depth (in the active layer). These temperatures are significantly colder than at Gruben. Occurrences of buried massive ice with thermokarst ponds are documented by visual observations and core drilling (Ugalde et al., 2018). The landforms 1, 2 and 3 can clearly be defined to be "rock glaciers" with steep fronts/lateral margins and recognizable ridge-and-furrow surface morphology, while the large debris mass marked with the numbers 4 and 5 neither shows clear characteristics of the landform "rock glacier" nor of the landform "debris-covered glacier". Here, thermokarst ponds develop in remains of massive surface ice from perennial ice patches/avalanche cones and small glacierets on top of

creeping and hydraulically impermeable, perennially frozen materials in deep-reaching permafrost. Three primary conclusions can be drawn from this example:

- Complex or doubtful landforms with combinations of surface ice and frozen ground represent conditions in nature that vary beyond limits of "either-or"-type landform categorization.
- Definition and measurement of material properties, ground thermal regimes and physical processes can provide more differentiated insights than applying simple landform schemes.
- Creep phenomena in mountain permafrost are more widespread than the occurrence of specific rock glacier landforms.

## 8 … and its effects on ice loss as a response to long-term warming trends

Ice melting and ground thawing processes are again fundamentally different in thermally controlled permafrost as compared to debris-covered massive ice (Figure 3; cf. Haeberli and Vonder Mühll, 1996). As a response to surface warming caused by changes in time of atmospheric conditions, by creeping out of shadows towards warmer areas in the sun and/or at lower altitudes, or by a combination of both, active layer thickness on rock glacier permafrost increases and thereby adjusts to (near-) equilibrium conditions (cf. long-term measurement series reported by PERMOS, 2023). Thawing into the *perennially frozen ice-rock mixture* underneath sets free the necessary debris for active-layer thickening. Thaw settlement primarily results from melting of excess-ice within the frozen debris, i.e., the ice occupying space within the frozen subsurface matter that exceeds the naturally available pore space, while melting of ice within its pore space alone would not lead to settlement. This process coupling is among the principal reasons for the remarkably low subsidence rates typically observed during recent decades on rock glacier permafrost (Fey and Krainer, 2018; Vivero et al., 2021; Gärtner-Roer et al., 2022). Latent heat effects (energy required to melt ice), together with the increasing thermal protection of permafrost as active layer thickness increases, make the response to climate change through degradation of ice-rich permafrost extremely slow. Further, a thickening of the coarse active layer has a substantial impact on the heat transfer between the atmosphere and the permafrost. First, the thermal resistance of the active layer increases as the thickness of the typically unsaturated debris layer increases. Air has a much lower thermal conductivity than ice or frozen ground (e.g., Andersland and Ladanyi, 2003; Arenson et al., 2021), which is why the thermal conductivity of the active layer tends to decrease as a result of permafrost degradation, contrasting the cover of a debris-covered glacier that cannot change its thermal resistance over time. Secondly, and potentially more importantly, a thickening of the dry and coarse active layer allows increased air flow and with that additional cooling through air convection (Wicky and Hauck, 2020). The Rayleigh number, which describes the potential and the strength of natural convection in porous media (Kane et al., 2001; Nield and Bejan, 2017), is directly dependent on the thickness of the active layer. As illustrated in Figure 3, natural convection can increase, or start to form over time in thickening active layers of degrading rock-glacier permafrost, but remains unchanged for a debris covered glacier.

The debris cover on *glaciers or buried massive ice* tends to be much thinner than the active layer on permafrost (Figure 4, cf. McCarthy et al., 2022), providing much smaller insulation capacity and potential for cooling effects by air convection. Most importantly, however, it can generally not significantly increase in thickness as a consequence of ice melting alone, because no significant mass of debris is provided from the vanishing of the underlaying massive ice. Assuming 0.1% average debris content inside the ice as an upper-end value based on Bozhinskiy et al. (1986), Nakawo et al. (1986), Kirkbride

and Deline (2013), Miles et al. (2021) or Anderson et al. (2021), the melting of a 100m column of ice would result in a thickening of surface debris by only 10cm. Continued warming therefore unavoidably leads to marked thermal imbalance and irreversible ice melting with enhanced surface subsidence until complete vanishing of the covered massive ice. The melting of ice thereby has a heterogenous spatial pattern as it directly responds to the highly variable thickness of the debris cover with its spatially heterogenous sources from rock falls, debris flows or various local displacements, furthered through ice melting

caused by thermal and mechanical erosion through water flow at the ice surface (Iwata et al., 1980; Mölg et al., 2020). This is in strong contrast with the thermally controlled active layer of rock glacier permafrost, which responds to climatic conditions in a much more homogeneous way. As a logical consequence, for which more measured evidence is still needed, bodies of massive ice buried in rock glacier permafrost are expected to disappear more quickly in response to warming surface conditions than surrounding frozen debris which can continue to exist over extended time periods (centuries, millennia). This may be the

primary reason why such massive buried ice seems to primarily occur, if at all, in upper reaches of presently existing rock glaciers (Haeberli and Vonder Mühll 1996; cf. Bosson and Lambiel, 2016) and hardly ever appears at advancing fronts.

The here-described material-process coupling causes the striking difference in the response of rock glaciers and debris-covered glaciers to ongoing atmospheric warming trends. It constitutes another reason why debris-covered glaciers have no choice – they can be in contact with creeping frozen materials but cannot, by themselves, become rock glaciers. They

remain debris-covered glaciers even under permafrost conditions with cold or polythermal ice, and can quite easily be discriminated from the remarkable viscous creep features usually called rock glaciers. The technical guidelines developed by RGIK (2023) rightly state that confusion between the two phenomena must be avoided. This is of particular importance when assessing and projecting how different components of the cryosphere respond to climate change and in the context of their hydrological role within a watershed.

**9 Summary and recommendation**

A combination of striking morphological, thermal and dynamic characteristics makes the difference between rock glaciers as viscous creep features in mountain permafrost and debris-covered glaciers (and smaller forms of surface ice) under conditions of ongoing global warming: convex versus concave shape, sharp versus diffuse edges, structured versus chaotic surfaces, continued coherent flow and advance versus slowing-down, disintegration and down-wasting. The test at Gruben and Yerba

Loca illustrates the applicability of such criteria in concrete climate-related inventory and monitoring work and confirms limits and complexities of landform interpretation needing further exploration.

The rich available quantitative knowledge basis from borehole and geophysical data in combination with advanced material-/process-related understanding enables safe and straightforward discrimination between rock glaciers as viscous creep phenomena in ice-rich mountain permafrost and debris-covered glaciers. The corresponding strategies recommended by experts of the International Permafrost Association are informed by and developed based on the process understanding and rich quantitative knowledge basis from numerous sophisticated field investigations using advanced technologies. They are clear and easy to follow, and may be especially helpful in cases when inventories are being compiled without comprehensive site investigations including geophysical soundings or boreholes. The treatment within internationally coordinated global climate observation of thermally controlled subsurface ice in rock glaciers as part of the Terrestrial Network for Permafrost (GTN-P) and of surface ice in debris-covered glaciers as part of the Terrestrial Network for Glaciers (GTN-G) is fully appropriate and such a differentiation shall be followed into the future.

Complex contact zones of surface ice (mostly thin perennial snow and ice patches, glacierets or small glaciers) with creep phenomena in ice-rich permafrost, however, in cases constitute diffuse landforms beyond "either-or"- type landform classification. Investigating the thereby involved relations and interactions is a challenging research field beyond attribution of simplistic "origins" to landforms. Exploring contacts and combinations of surface and subsurface ice with their strikingly different response characteristics concerning atmospheric warming is now indeed a growing field of advanced research. It involves quantitative treatment of the involved material properties and processes. This by far exceeds the possibilities of speculative interpretations based alone on visual surface inspection. A recent example illustrating the potential of multimethod field measurements to be used in such complex cases is the comprehensive investigation at the Chauvet site in the French Alps (Cusicanqui et al. 2023).

The key physical phenomenon relates to the fundamentally different distribution of ice and debris: subsurface frozen ice-rock mixtures in rock glaciers versus massive ice with debris on top in debris-covered surface ice. The related mechanical coupling between the moving body at depth and the surface layer of debris is tight in the case of perennially frozen rock glaciers but virtually inexistent in the case of surface ice with a debris cover. The strikingly different morphologies of advancing fronts of rock glaciers and of debris-covered glaciers result from this difference and are in most cases easily recognizable. The difference in ice-debris distribution also explains the extreme contrast in reactions to global warming: Slow thaw subsidence, increasing creep rates and continued advance in the case of warmed-up and softened frozen debris/talus in rock glaciers versus down-wasting, slowing down and disintegration/collapse/vanishing of debris-covered glaciers.

*Data availability*: The resistivity raw data can be received via e-mail from Julie Wee (julie.wee@unifr.ch), who is currently preparing a publication "Characterizing ground ice content and origin to better understand the seasonal surface dynamics of the Gruben rock glacier (western Swiss Alps)" which is planned to be submitted to The Cryosphere in early 2024, where all geophysical data from this site will be published together.


*Team list and author contribution:* WH and LA developed the original concept and prepared a first draft. JW and CH provided information about the most recent field campaign at the Gruben site. NM specifically helped with covering aspects related to debris-covered glaciers. All co-authors were actively involved in the preparation, discussion, revision and finalization of the submitted version.

*Competing interests.* At least one of the (co-)authors is a member of the editorial board of The Cryosphere.


**Acknowledgements**

We appreciate and support the ongoing efforts undertaken by the International Permafrost Association (IPA) to compile systematic and worldwide inventories of viscous creep features in mountain permafrost – the so-called "rock glaciers" with their warming-related kinematics. The intention of the present contribution is to publicly and critically reflect and strengthen the data- and physics-based concepts thereby developed and their application within the framework of the Global Terrestrial Network for Permafrost (GTN-P/GCOS) as part of integrated global cryosphere observation. We especially thank the numerous colleagues who continue contributing to this important work by documenting solid facts using advanced technologies and process understanding. An anonymous reviewer and Adriano Ribolini provided constructive feedback to our contribution.

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

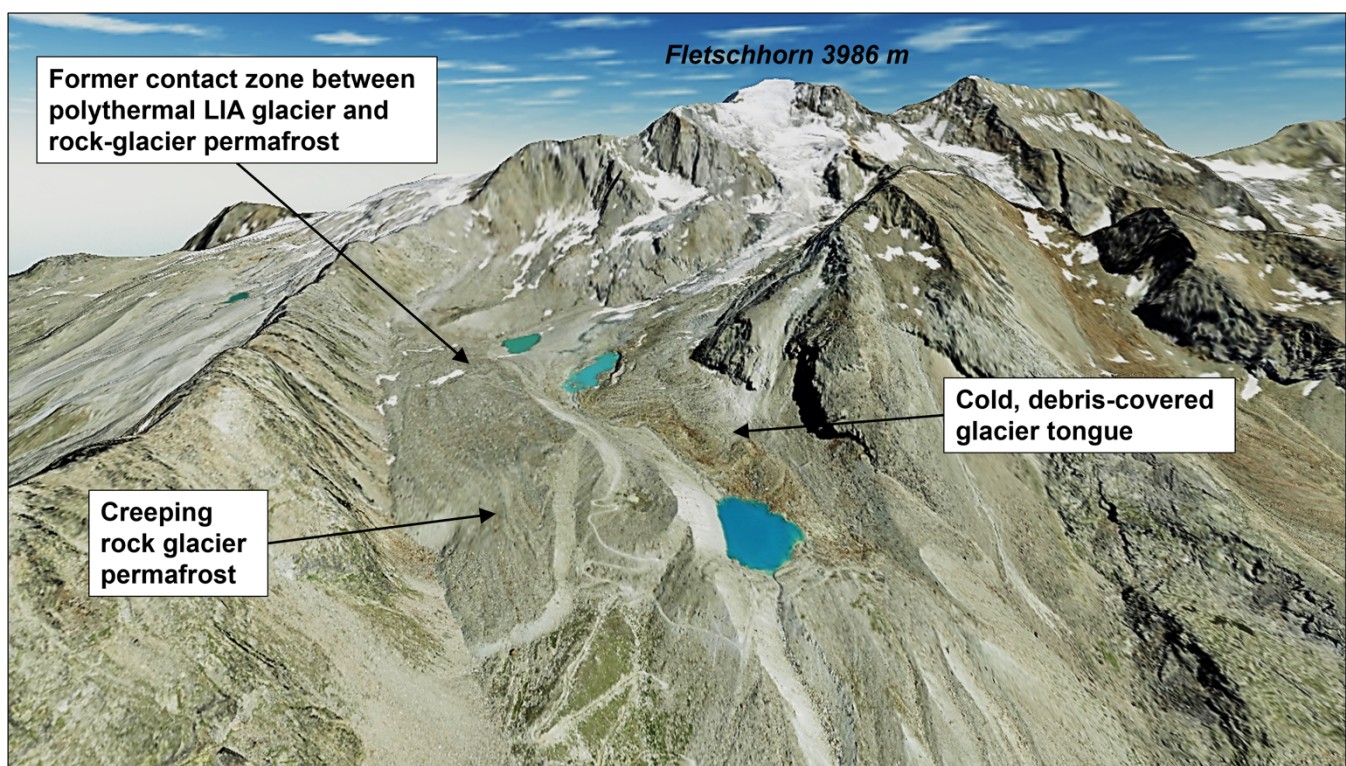

**Figure 1: Continued viscous creep of perennially frozen debris, rich in ground ice at Gruben rock glacier (left), decaying debris-covered, cold tongue of Gruben glacier (right), and former contact zone (center) between the polythermal Little Ice Age (LIA) glacier and the creeping permafrost of the Gruben rock glacier. World Imagery available from Global Mapper V23.1; recent, but exact date unknown.**


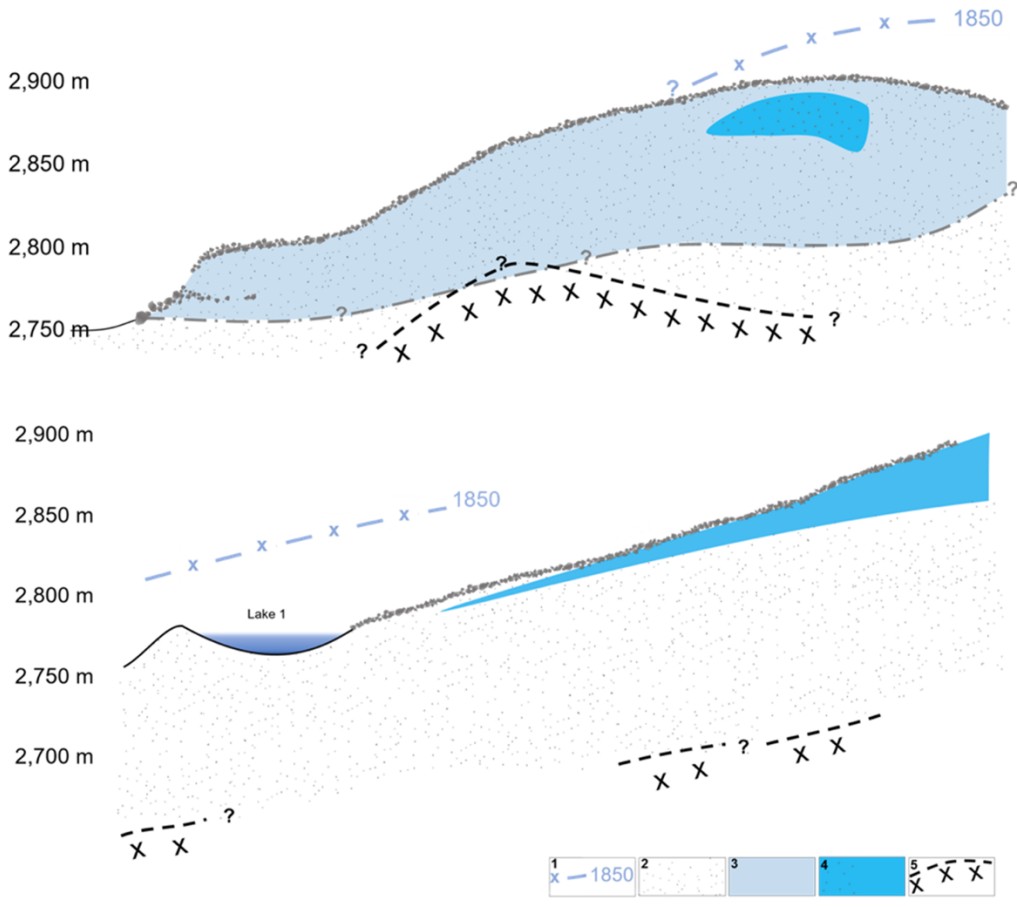

**Figure 2: Schematic longitudinal profile through the Gruben rock glacier (top) and the cold debris-covered tongue of Gruben glacier (bottom) based on extensive geophysical prospection (electrical resistivity, seismic refraction, radio-echo-sounding as compiled by Haeberli et al., 2001; Gärtner-Roer et al., 2022) and on recent soundings by Wee et al. (2022). The upper part of the rock glacier presentation reflects the complex contact zone between the polythermal LIA Gruben glacier and the rock glacier permafrost (Kääb et al., 1997); flow direction in this zone is not parallel to the profile but somewhat towards the viewer, leading to the orographic right LIA moraine of Gruben glacier rather than to the advancing rock glacier front. Quantitative information about the debris-covered part of Gruben glacier is from a combination of earlier glacier-bed resistivity and radio echo soundings (Haeberli and Fisch, 1984) with measurements of long-term changes in surface elevation as reported by Gärtner-Roer et al. (2022). Vertical exaggeration is by a factor of 2. Legend: (1) maximum Little Ice Age extent of Gruben glacier; (2) talus, debris; (3) permafrost conditions; (4) buried massive ice of uncertain origin (Gruben rock glacier) and glacier ice (debris-covered part of Gruben glacier); and (5) bedrock after unpublished seismic refraction (Gruben rock glacier) and glacier-bed resistivity soundings (Gruben glacier).**

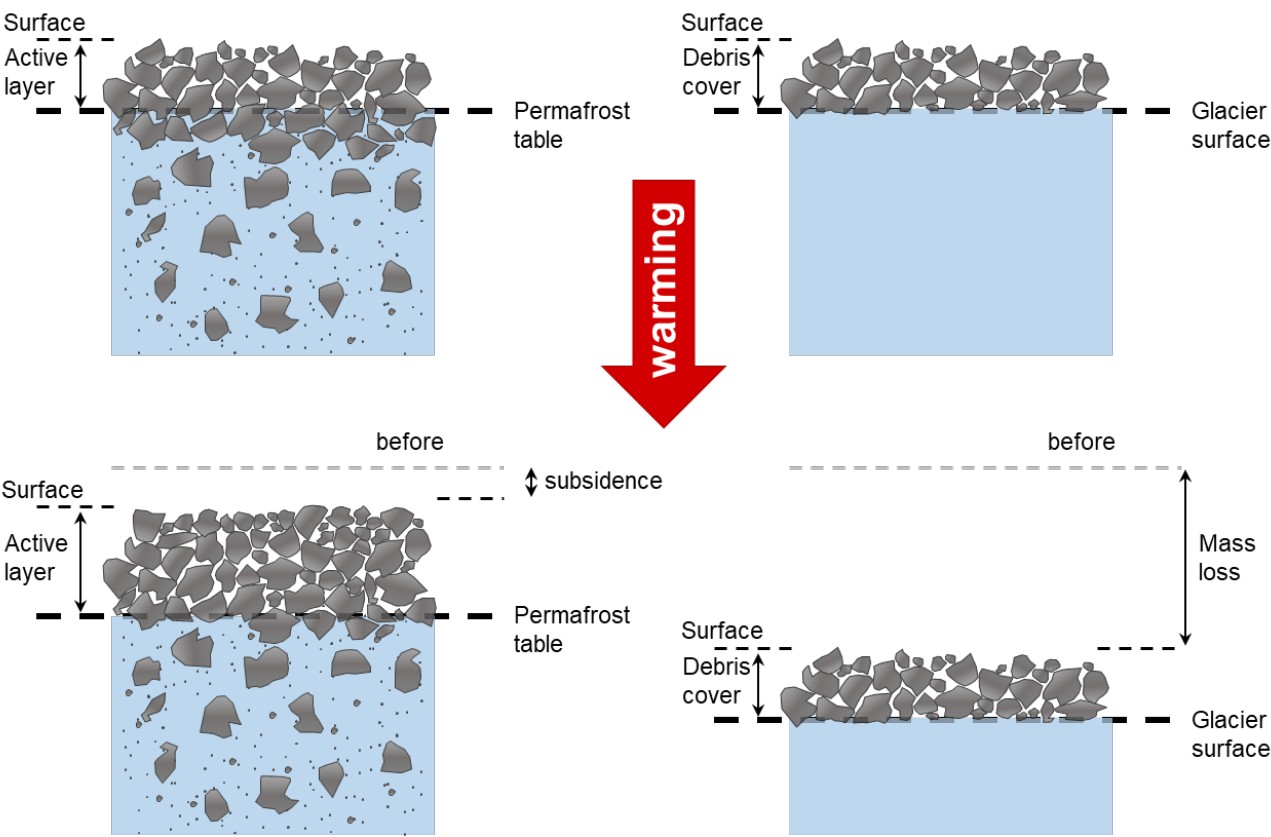


**Figure 3: Types of ice-debris coupling and adjustment to warming effects in the cases of perennially frozen debris with excess ice (left), and debris-covered massive ice (right). As the ice melts, the thermal insulation in the form of the active layer increases for the permafrost case, but remains unchanged in the case of a debris-covered glacier, where the debris thickness largely remains constant.**
**Modified from Haeberli and Vonder Mühll (1996).**

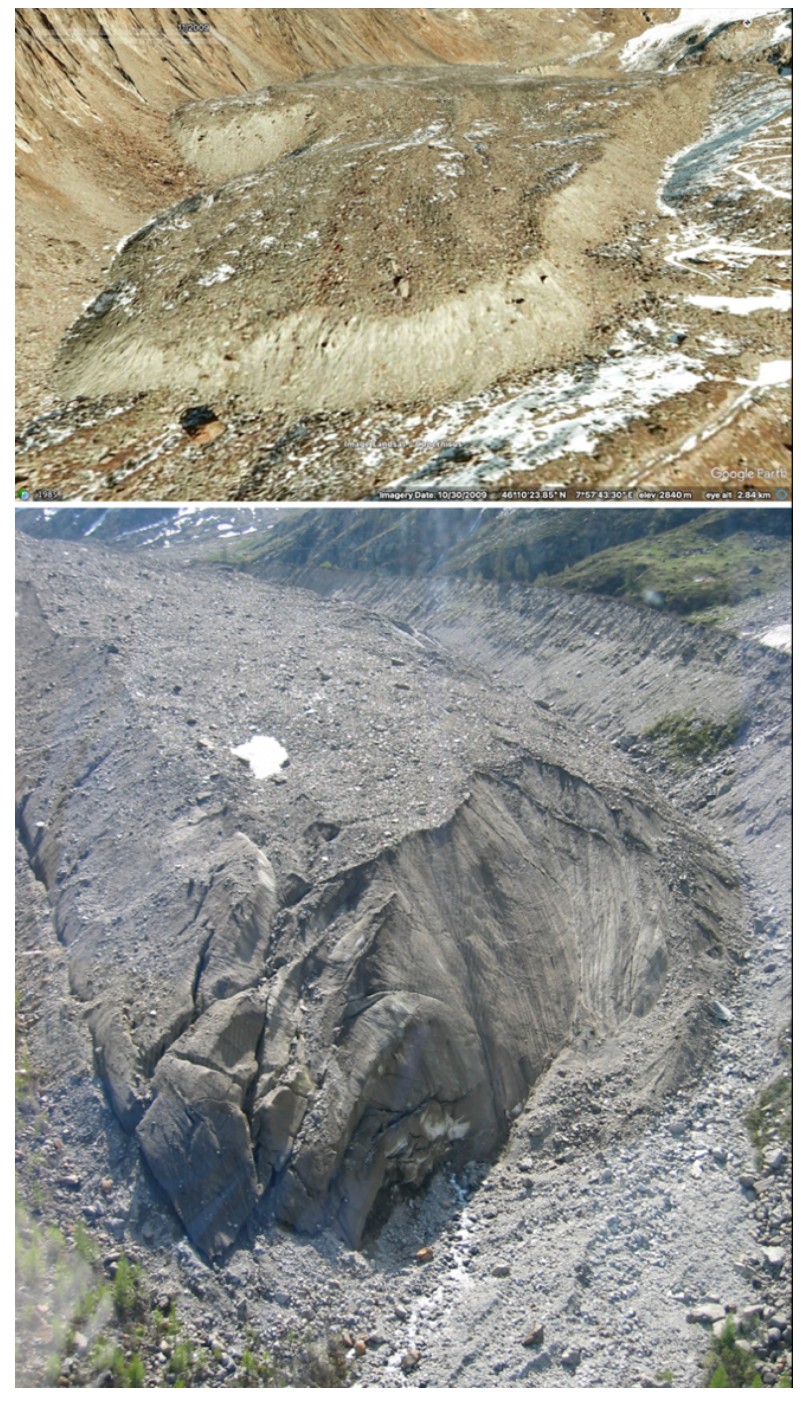

Figure 4: Advancing fronts of creeping perennially frozen debris at Gruben rock glacier (top; fall 2009, © Google Earth) and of the debris-covered Belvedere glacier in the Italian Alps during its intermittent advance around the turn of the century (bottom; summer 2005).

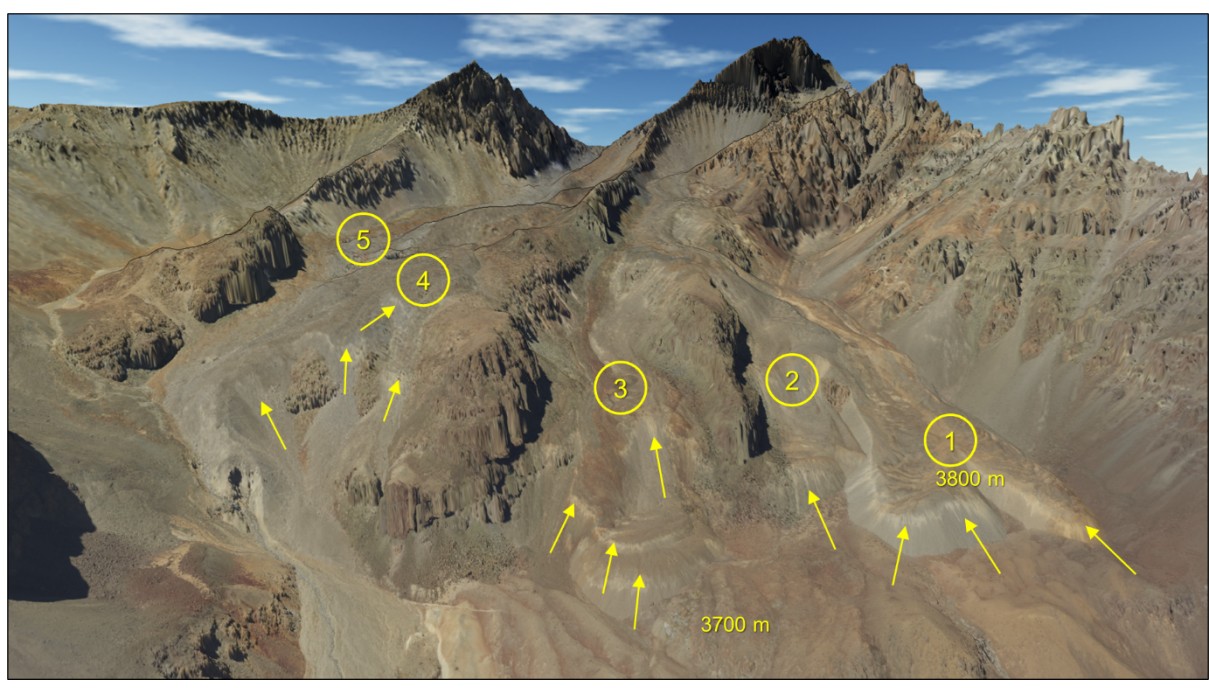

**Figure 5: Creep phenomena in the upper Yerba Loca Valley, Andes of Chile. Yellow arrows point to over-steepened advancing fronts of creeping perennially frozen talus/debris, numbers in circles point to landforms as discussed in the text. © DigitalGlobe available in GlobalMapper, modified.**