# Peer review of "Discriminating viscous creep features (rock glaciers) in mountain permafrost from debris-covered glaciers – a commented test at the Gruben and Yerba Loca sites, Swiss Alps and Chilean Andes"

_EGUsphere, 2023_

## Referee Comment (RC1)

[RG - rock glaciers, DG - debris covered glaciers]

**Summary**
This invited perspective manuscript discusses the application of criteria for identifying/detecting rock glaciers for two example sites, Gruben and Yerba Loca. A main focus is given to differentiating RG from DG. Following the introduction, Section 2 details how the criteria are applied to the Gruben site to distinguish the Gruben rock glacier from the neighbouring debris covered glacier. Section 3 discusses more complicated cases (contact zones of surface and subsurface ice) and concludes that *"neither the term "rock glacier" nor the term "debris-covered glacier" would be appropriate for such complex contact zones with their characteristically diffuse landforms."*

Section 4 critically states that contrasting views regarding the definition and genesis of rock glaciers exist in the scientific community, particularly regarding how RG differ and should be differentiated from DG. This section functions as a starting point for the following sections, which compare and contrast specific characteristics of RG and DG: Section 5 highlights that RG move as a result of viscous creep of permafrost. The "coherent movement pattern" of the creeping permafrost is transmitted to the surface debris, which is "interlocked" with the deeper layers. This results in characteristic furrows and ridges. In contrast, the surface debris of DG is not "interlocked" with the ice and there are no "coherent flow patterns" on the surface. Section 6 highlights that RG have oversteepened advancing fronts while advancing DG often show massive ice at the terminus. The Yerba Loca site is used as an example illustrating oversteepened fronts at RG and other features of creeping permafrost. Section 7 focuses on differences in the response of RG and DG to climate warming.

Section 8 reiterates key points made in the introduction and throughout the manuscript and concludes that RG and DG should and usually can be clearly distinguished based on the strategies developed by the IPA. Further, as stated in the introduction, the authors recommend that RG as permafrost phenomena remain under the purview of the GTN-P, while DG remain part of the GTN-G.

**General comments**
This is an important contribution to the ongoing discussion on identifying rock glaciers, e.g. for inventory purposes, and distinguishing them from debris covered glaciers and other landforms. A consistent approach and clear definitions are needed for RG inventorization and the RGIK/IPA guidelines provide a broadly applicable supporting framework for such efforts, many of which are currently ongoing in mountain regions around the world. The manuscript makes valuable points about separating RG from DG and will further support the development of community strategies and guidelines in this area. I am sure my comments/questions can be resolved and look forward to seeing this published in TC.

Terminology and definitions:
In my opinion it would be beneficial to explicitly include the wording of the guidelines/criteria that are applied, tested and referred to throughout, as well as clear definitions of RG and other terms. Perhaps the introduction could be expanded by a short dictionary style section listing key terminology. This could also include the brief comment on subsurface and surface ice currently in the supplement. The authors point out repeatedly that DG remain DG no matter what, given that they do not turn into RG (a central message of this manuscript). Accordingly, small dead ice bodies that can remain when glaciers melt and may be partially or completely debris covered are still considered DG ("glaciers") in the sense that they are "not rock glaciers". However, they are also no longer "glaciers" in the typical sense. The authors sometimes use the more descriptive phrase "debris covered surface ice" or similar for such cases, but the usage is not always consistent. Defining "glacier" in some way or always using variations of "surface ice" in the context of "complex cases" could help prevent terminology related confusion. The authors also mention a size limit that separates glaciers from "not glaciers" in glacier inventories. It would be helpful to state what this limit is. The readership of TC can certainly make an educated guess about this and other matters, but the clarity of the manuscript could nonetheless be improved by adding some definitions and consistently using the respective terminology.

Complex cases, ambiguous landforms:
The authors state in the introduction:

*"An objective way of differentiating corresponding landforms and kinematics is essential in creating clarity when utilizing such landforms to assess where and how climate change impacts our planet, specifically the cryosphere, or when used in a regulatory/legal context, for example in view of hydrological significance; or generally, to avoid confusion and duplication."*

It is certainly important to distinguish RG from DG for these purposes. It is also important to have a practical and consistent way of dealing with ambiguous landforms that cannot easily be categorised as either DG or RG particularly for inventorization purposes. When inventorization of cryospheric features is connected to regulatory measures, the measures in question often do not themselves give clear definitions of what exactly should and should not be included in the inventory. Having a community consensus on landform definitions beyond RG and DG and on dealing with ambiguity would be beneficial in such cases. Do the authors have a recommendation or further comments on how to deal with ambiguous landforms (buried surface ice as well subsurface ice other than RG) from an inventory perspective, or how a consensus based community strategy to this end might be developed?

I do not fully understand what the authors are suggesting related to the application of a size threshold and would appreciate more detail on this. Is the idea that a threshold (smaller than glaciers, thinner than underlying permafrost) helps identify the "complex cases" in a general sense, or should such thresholds be used as additional criteria alongside those of the RGIK/IPA for classification purposes? If the latter, how would this size limit be applied practically, for example when compiling an RG inventory? Should the excluded features be ignored in inventorization efforts?

Contrasting views:
The authors point out work by others that presents somewhat differing views on the distinction between RG and DG and RG genesis. Section 4 in particular is critical of these works. The authors have strong arguments that can stand alone and are not further strengthened by dismissive comments towards others. I would suggest revisiting this section and either expanding the overview of contrasting work for a more comprehensive picture (e.g. Knight et al, 2019; Jones et al, 2019, and others) or finding a more concise way of introducing the following sections.

**Specific comments:**
**Introduction:**
For clarity and to help the reader, I suggest including:

- the specific "proposed technical recommendations/guidelines" that are to be tested. Since the stated aim of the manuscript is to test and comment on the guidelines, it would be useful to explicitly mention what these guidelines are.
- the "technical definition of rock glaciers'' used by the RGIK/IPA, assuming the authors agree with this definition. A reference to section 4 of the manuscript's supplement (surface vs subsurface ice) could be added alongside the definition of rock glaciers to set the stage for the surface/subsurface arguments that follow in the later sections. Alternatively, the short paragraph explaining this as currently contained in the supplement could simply be added to the main text.

The two RGIK documents cited by the authors list two mandatory "geomorphological criteria" for rock glacier detection (front and lateral margins) and one optional criterion (ridge-and-furrow topography, section 3a in the RGIK (2022) baseline concepts). The manuscript discusses RG fronts (Section 6) and ridge-furrow topography (Section 5) but does not mention the lateral margins criterion. If the aim is to test the RGIK criteria, a brief explanation of why lateral margins as a mandatory criterion are being excluded from this exercise of testing and commenting would be helpful. If criteria other than those listed in the RGIK document are being tested, please clarify.

**Section 2: Rock glacier and cold debris-covered glacier at the Gruben site**
L95 *The differences are obvious, and the morphological criteria proposed by the IPA action group are adequate*
Consider specifying what these criteria are.
Fig 2: I suggest adding a reference to the very helpful Fig. Sup.-1 in the supplement to the caption of this figure.

**Section 3: Complex zones with contacts between surface and subsurface ice**
L 125 *not every piece of surface ice is a "glacier"*
It might be useful to briefly define your usage of the term "glacier" here or earlier in the manuscript, see general comment.

L 139 *mostly smaller than the lower size limit applied to the term "glacier" in glacier inventories*
For clarity, please state what this size limit is / give citations. Not all glacier inventories use consistent size limits and some make a case for including very small, debris covered ice bodies in regions where deglaciation is imminent. (See e.g. the discussion in Section 5.4 of Fischer et al (2021))

**Section 4 - debris covered glaciers remain debris covered glaciers**
L 148: What would the authors consider "adequate" in this context? Perhaps a statement could be made on minimum required quantitative information.

L151: can DG turn into "complex cases" of buried surface ice in contact with permafrost features as discussed in the previous section as a third option? If so, should that be added to the list as option C? If not it would be useful to briefly clarify.

L153 *lavastream-like*
Consider omitting in the interest of precise usage of terminology.

L163 *Nevertheless, it is useful to understand the reason why this is the case …*
Why what is the case? I find this sentence hard to follow, consider rephrasing. See also the general comment.

**Section 6 … as visible at advancing fronts…**
L217: *Advancing debris-covered glaciers have become exceptional under conditions of atmospheric temperature rise and predominating glacier shrinkage. In such increasingly rare cases, massive ice of the flowing glacier is usually visible at near- vertical fronts where debris cannot accumulate (Figure 4).*
Does this imply that Belvedere Glacier (Fig 4) is currently advancing? Afaik that is not the case.

L220: *The advantages of adequately interpreting over-steepened fronts of creeping frozen talus/debris can be illustrated with an example from the Andes….*
Fig 5 shows oversteepened fronts marked by yellow arrows at rock glaciers (1, 2, 3) and ambiguous landforms (4, 5). What information is gained by "adequately interpreting" the oversteepened fronts and what does "adequate" mean in this context? Perhaps it would be informative to walk through all three of the RGIK/IPA criteria for rock glacier detection (fronts, lateral margins, furrow-ridge topography) for the Yerba Loca landforms to clearly show how the criteria can help separate rock glaciers from other landforms.

**Section 7 … and its effects on ice loss as a response to long-term warming trends**
L267: *…and hardly ever appears at advancing fronts*
Are there studies showing massive ice at advancing fronts? citations?

**Section 8**
L277 *The rich available quantitative knowledge basis from borehole and geophysical data in combination with advanced material- /process-related understanding enables safe and straightforward discrimination between rock glaciers as viscous creep phenomena in ice-rich mountain permafrost and debris-covered glaciers. The corresponding strategies recommended by experts of the International Permafrost Association are clear and easy to follow.*
I suggest briefly stating again what the recommended strategies are. Borehole and geophysical data are available only for a small fraction of RG, DG, and other landforms. It could be pointed out that since the RGIK/IPA strategies are informed by and developed based on the process understanding and rich quantitative knowledge basis the

authors refer to, they may be especially helpful in cases when inventories are being compiled without comprehensive geophysical information or boreholes.

L283 *Complex contact zones of surface ice (mostly perennial snow and ice patches, glacierets or small glaciers) with creep phenomena in ice-rich permafrost in cases constitute diffuse landforms beyond "either-or"- type landform classification.*

I understand that an in depth discussion may be beyond the scope of this manuscript, but would the authors consider these ambiguous landforms the responsibility of the GTN-P or GTN-G? How should they be inventoried? I agree that they are "beyond simplistic landform attribution" but they are relevant for inventories as assessments of the changing cryosphere in regulatory and or hydrological contexts.

Personally, I would like to see the conclusion link back to the stated aim of the introduction, i.e. testing the application of the guidelines of the RGIK/IPA at the Gruben and Yerba Loca sites. Perhaps a brief summary pertaining to this aspect of the manuscript could be added, maybe with some generalised conclusions regarding the usefulness of the specific mandatory and optional criteria (as per RGIK 2022, 3a) that can be derived from the characteristics of the two test sites, i.e., RG and DG in close vicinity at Gruben and different kinds of creeping permafrost at Yerba Loca.

**Typos and such:**

L161 &  L271: I  suggest replacing "safely" with "definitively" or a similar word.
L261: Check spelling of Moelg/Mölg in citation

**Supplement:**
Part 2, kinematics Yerba Loca: Can you indicate which of the numbered features (1-5) in Fig 5 are shown in Fig.-Sup. 1-3?
Fig. Sup-2: typos in the caption, "Sub" instead of Sup
Part 4: not referenced in the manuscript? I suggest moving this f

**References:**
Fischer, A., Schwaizer, G., Seiser, B., Helfricht, K., & Stocker-Waldhuber, M. (2021). High-resolution inventory to capture glacier disintegration in the Austrian Silvretta. *The Cryosphere*, *15*(10), 4637-4654.

Jones, D. B., Harrison, S., & Anderson, K. (2019). Mountain glacier-to-rock glacier transition. *Global and Planetary Change*, *181*, 102999.

Knight, J., Harrison, S., & Jones, D. B. (2019). Rock glaciers and the geomorphological evolution of deglacierizing mountains. *Geomorphology*, *324*, 14-24.

RGIK (2022). Towards standard guidelines for inventorying rock glaciers: Baseline concepts (version 4.2.2). IPA Action Group Rock glacier inventories and kinematics, 13 p. https://bigweb.unifr.ch/Science/Geosciences/Geomorphology/Pub/Website/IPA/CurrentVersion/Current_Baseline_Concepts_Inventorying_Rock_Glaciers.pdf

---

## Author Comment (AC1)

**On rock-glacier permafrost in global climate observation – Reply to community comments**

We thank Stephan Harrison and Brian Whalley/Fethi Azizi for their Community Comments. Their texts document their personal ideas, beliefs and opinions concerning rock-glacier origins. Their argumentation is essentially based on intuitive landform interpretation from visual surface inspection and in view of "airing competing (and sometimes contradictory) views in science", as Harrison formulates it.

Our contribution relates to internationally coordinated permafrost and glacier programs for UN/ICSU-related global climate-system observation that are based on consensus from multiple rock glacier researchers. Within this policy-relevant framework, science has the responsibility to strictly relate to measured facts about the involved conditions, materials, processes and time scales. Referring to the measured evidence provided in our contribution and based on the advanced state of quantitative knowledge available in the modern literature about permafrost and glaciers in cold mountain regions (cf. the references in our article), we here briefly repeat and summarise some most essential points.

1. The long-term existence of subsurface ice in rock glaciers first of all relates to thermal conditions. To define such thermal conditions is, therefore, a basic requirement for climate-related research and observation of permafrost and rock glaciers. Quantitative information is obtained from climate data, miniature temperature logging, borehole measurements or numerical model calculations (e.g., Haq and Baral 2019, Baral and Haq 2020, Li et al. 2023), optimally in combination, and if possible supported by geodetic measurements of flow characteristics to define activity levels (cf. Bertone et al. 2023). The results are clear, especially also in view of specifically cold microclimatic conditions (mountain shadow, ventilated blocks at the surface, long-lasting seasonal snow), which characterise most rock glaciers: active, ice-containing rock glaciers occur where mean annual temperatures are typically negative or have until recently been negative. The technical term permafrost defines this specific geothermal condition (negative subsurface temperature throughout the year; Muller 1947). With other words: Ice-containing rock glaciers are – independently of any material characteristics – thermally in a permafrost condition.

2. As a consequence, the question "is it permafrost or is it a glacier?" as it seems to be central in the Whalley/Azizi comment and in the theoretical concept of "equifinality" of rock glacier origins as maintained by Harrison, is inconsistent. The real question is whether and, if yes, under what conditions and to what extent various forms of buried surface ice can be part of mountain permafrost.

3. Negative subsurface temperatures in nature induce freezing processes. In the case of rock glacier permafrost, freezing processes are forced by negative sub-zero temperatures typically lasting over millennial time scales as documented by numerous absolute age determinations. Such slow and long-term freezing at depth produces large amounts of ice inside the affected rock material. An increasing number of core drillings and borehole geophysics, together with hundreds of sophisticated geophysical soundings definitely document that rock-glacier permafrost consists of perennially frozen ice-rock mixtures with characteristic electrical resistivities of tens to hundreds of k$\Omega$m (Herring et al., 2023). Ice contents by volume are locally variable but mostly far larger – on average by about a factor of two – than the pore volume of

the original rock material under unfrozen conditions (Hausmann et al. 2007; Monnier and Kinnard 2015). Small and large lenses of massive ice from ice segregation in frost-susceptible materials (silts, fine sands) are common.

4. Mostly thin and isolated remains of surface ice with extreme electrical resistivities (MΩm) can in cases be embedded in these ice-rich frozen sediments, primarily in upper parts of rock glaciers. They are clearly the exception rather than the rule. The belief that rock glaciers could entirely be debris-covered Little Ice Age glaciers has long been disproved by drillings, geophysical soundings and absolute age dating. It is difficult to understand for us, how the Whalley/Azizi comment can maintain such an unrealistic idea against the rich quantitative information available for decades already from modern research on mountain permafrost.

5. The development by long-term freezing processes of excess ice or ice-supersaturation inside the affected rock material induces fundamental changes in the material properties and geotechnical characteristics of originally non-cohesive talus/debris material with high internal friction. The rheology of such materials in perennially frozen, ice-supersaturated condition with high cohesion and reduced internal friction constitutes a classical topic in permafrost engineering. It is generally defined as a function of applied stress, temperature and ice content, all of which vary spatially. Unfrozen water contents, which depend on the unfrozen water characteristic of the sediments present in the rock glacier (mainly driven by the fines content), increase at near-thawing temperatures and thereby certainly induce important effects (including a reduction in effective stress). Such characteristics help with explaining the documented ongoing, warming-induced increase of creep rates in rock-glacier permafrost. Again, it is difficult to understand for us how the Whalley/Azizi comment can continue to deny this long-established quantitative knowledge and understanding.

6. The focus and primary interest of climate-related glacier and permafrost observation in cold mountains concerns the ongoing dynamic evolution. Using modern geodetic/remote-sensing methods, coherent flow fields and their changes in time can be determined with high precision and over large areas, including remote sites (cf. Figures Sup.-3 and Sup.-4). The obtained results can be compared with measured or numerically modelled changes in thermal conditions. Related inventory work concerning rock glaciers and (debris-covered) glaciers should focus on landforms, which are clearly recognizable as such and which can be discriminated from each other. The permafrost and glacier networks (GTN-P and GTN-G) of GCOS/GTOS under the umbrella of international scientific associations (IPA, IACS) are well organized for such work. Complex contact zones of surface ice and permafrost, however, are beyond straightforward "either-or" landform schemes, should neither be included in permafrost nor glacier inventories, but need further exploration.

7. Exploring contacts and combinations of surface and subsurface ice with their strikingly different response characteristics concerning atmospheric warming is indeed a growing field of advanced research. It involves quantitative treatment of material properties and processes. This by far exceeds the possibilities of speculative interpretations based alone on visual surface inspection. A recent example illustrating the potential of multimethod field measurements to be used in such complex cases is the comprehensive investigation at the Chauvet site in the French Alps (Cusicanqui et al. 2023).

*Specific remarks on the Whalley/Azizi comment:*

- Gruben glacier is not "indicated as a 'cold, debris-covered glacier tongue' with no evidence for its temperature regime". Gruben glacier is documented to be polythermal based on published borehole temperatures, radio-echo soundings and glacier-bed resistivities as explained in our contribution.

- Gruben rock glacier is not "interpreted … as a permafrost body because of the low surface velocity (<1m/a); a kinematic explanation". Gruben rock glacier is documented (a) to be in permafrost condition based on measured temperatures combined with spatial permafrost mapping/numerical modelling, and (b) to consist of perennially frozen debris based on comprehensive geophysical soundings and core drilling; few, thin and isolated remains of buried surface ice only exist in the former contact zone with the Little Ice Age glacier.

- Melt pools preferentially form in massive buried ice, especially where those are embedded in permafrost. The reason is, that the low hydraulic permeability of ice-rich frozen ground underneath the melting buried ice helps keeping the water in the thermokarst depressions. The well-documented thermokarst lake at Gruben is a striking example. The generally limited areas and depths of melt pools indicates limited spatial extents and thicknesses of buried surface ice occurrences on rock glaciers.

- The statement "steep frontal RG slopes are not 'over steepened', they are at the appropriate resting angle of the granular material" is a fundamental misunderstanding, ignoring the involved dynamic processes and multiple field measurements. The fronts of actively advancing rock glaciers like Gruben rock glacier, and of distinct as well as indistinct creep features at Yerba Loca are constantly being over-steepened by the forward movement of the creeping frozen body with highest velocities at its top as documented by borehole deformation (cf. also Figure Sup.-3 in the supplement). The oversteepening in the upper part of the rock glacier front is caused by suction that forms in the unsaturated material, which may also be referred to as an apparent cohesion. Constant over-steepening is the reason why the upper parts of actively advancing fronts of rock glaciers and even of less distinct features of permafrost deformation remain unstable, causing continued rock falls, as the creeping body deforms, or the suction decreases, for example during a rainfall when water enters the ground. Such rock falls can include coarse blocks from the rock-glacier surface and build up the characteristic talus aprons at the foot of many fronts (cf. Figure Sup.-3) . This principle has already been explained by Wahrhaftig and Cox, references to modern precise measurements are found in our contribution. The constant oversteepening, destabilization and material-detachment process continuously exposes fresh material from the inside of the moving body in the upper parts of active rock glacier fronts. As illustrated with the Yerba Loca site in our contribution, the often strikingly bright appearance of freshly exposed rock material in upper frontal parts of actively advancing rock glaciers constitutes a characteristic and easily recognizable indication of continued creep in perennially frozen talus/debris. Luminescence dating documents that such freshly exposed materials had been underway inside the creeping body for thousands of years.

- The creeping material inside rock glaciers is not "partially to fully saturated rockfill". Such material indeed undergoes damped creep with finite deformation as documented, for instance, in the frozen ice-saturated blocks at depth underneath the main shear horizon of the Murtèl 1987 borehole. However, core drillings, borehole deformation measurements and numerous geophysical soundings again and again confirmed that long-term creep of rock glaciers takes place in highly supersaturated talus/debris, where the excess ice – in addition to strengthening cohesion – reduces internal friction (Springman et al. 2012). It is this ice-supersaturation, which enables secondary or steady-state creep with unlimited cumulative deformation. To put it in simple words: The frozen material of rock glaciers undergoing long-term creep is not "rocks with ice inclusions" as assumed in the Whalley/Azizi comment but "ice with rock inclusions" – geotechnically a fundamentally different material. The IPA/IACS task force report about permafrost creep and rock glacier dynamics includes a discussion of the mechanics involved (Haeberli et al. 2006; cf. also Arenson et al. 2021 concerning the physics of frozen ground/permafrost).

- This basic misunderstanding explains why the argument that "substantial (>20m) ice thicknesses are required for observable creep to occur in frozen debris, even on steep hillsides, particularly at 'low' temperatures" (Whalley and Azizi, 1994, 2003 as cited in the Whalley/Azizi comment) is not supported by field observations. Rock glaciers, for which borehole deformations are available (Arenson et al. 2002; Buchli et el. 2013; Fey and Krainer 2018) clearly demonstrate the presence of shear horizons where most of the observable deformation occurs. Such "shear horizons", which are actually zones of increased creep of the frozen material, can be less than 20 m deep as shown in these referenced publications. A simple, Glen-type flow law that does not consider the ground thermal regime of the permafrost and the complex geotechnical characteristics of typically heterogenous ice-rich frozen talus/debris materials, will just not reproduce the full range of observed deformation and velocity changes (e.g. Müller et al. 2016, Cicoira et al. 2021).

Personal note by Wilfried Haeberli: I am sorry for Louis Lliboutry to see his rude 1990 formulations being brought to the public again. In personal communication with me, the Journal of Glaciology had apologized for having published such wording. In my former function as Director of the World Glacier Monitoring Service (WGMS), I had – over several years – direct contacts and collaboration with Louis Lliboutry in his former function as president of the International Commission on Snow and Ice (ICSI; today IACS). I learned many things from this outstanding scientist, and I highly appreciated his always critical but nevertheless constructive and strong support of internationally coordinated glacier monitoring (comparable permafrost monitoring only started later). Lliboutry, on the other hand, changed his mind concerning rock glaciers and mountain permafrost after having read my 1985 publication and the first reports on the Murtèl core drilling. He even started encouraging younger colleagues in France to do similar quantitative field measurements on the topic. French scientists today are among the international leaders concerning research on mountain permafrost, rock glaciers and sites with complex contacts between surface and subsurface ice.

**References not contained in our contribution:**

Arenson, L., Hoelzle, M. and Springman, S.: Borehole deformation measurements and internal structure of some rock glaciers in Switzerland. Permafrost and Periglacial Processes, 13(2), 117-135. doi.org/10.1002/ppp.414, 2002.

Baral, P. and Haq, M.A.: Spatial prediction of permafrost occurrence in Sikkim Himalayas using logistic regression, random forests, support vector machines and neural networks. Geomorphology 371, 107331. doi.org/10.1016/j.geomorph.2020.107331, 2020.

Bertone, A., Jones, N., Mair, V., Scotti, R., Strozzi, T. and Brardinoni, F.: A climate-driven, altitudinal transition in rock glacier dynamics detected through integration of geomorphological mapping and InSAR-based kinematics. The Cryosphere Discussion. doi.org/10.5194/tc-2023-143, 2023.

Buchli, T., Merz, K., Zhou, X., Kinzelbach, W. and Springman, S. M.: Characterization and Monitoring of the Furggwanghorn Rock Glacier, Turtmann Valley, Switzerland: Results from 2010 to 2012. Vadose Zone Journal, 12(1). doi.org/10.2136/vzj2012.0067, 2013.

Cusicanqui, D., Bodin, X., Duvillard, P.-A., Schoeneich, P., Revil, A., Assier, A., Berthet, J., Peyron, M., Roudnitska, S. and Rabatel, A.: Glacier, permafrost and thermokarst interactions in Alpine terrain: Insights from seven decades of reconstructed dynamics of the Chauvet glacial and periglacial system (Southern French Alps). Earth Surface Processes and Landforms 48 (13), 2595-2612. doi.org/10.1002/esp.5650, 2023.

Hausmann, H., Krainer, K., Brückl, E. and Mostler, W.: Internal structure and ice content of Reichenkar rock glacier (Stubai Alps, Austria) assessed by geophysical investigations. Permafrost and Periglacial Processes, 18(4), 351-367, 2007.

Haq, M.A. and Baral, P.: Study of permafrost distribution in Sikkim Himalayas using Sentinel-2 satellite images and logistic regression modelling. Geomorphology, 333, 123-136. doi.org/10.1016/j.geomorph.2019.02.024, 2019.

Herring, T., Lewkowicz, A.G., Hauck, C., Hilbich, C., Mollaret, C., Oldenborger, G.A., Uhlemann, S., Farzamian, M., Calmels, F. and Scandroglio, R.: Best practices for using electrical resistivity tomography to investigate permafrost. Permafrost and Periglacial Processes, 34 (4), 494-512. doi.org/10.1002/ppp.2207, 2023.

Li, M., Yang, Y., Peng, Z., Liu, G.: Assessment of rock glaciers and their water storage in Guokalariju, Tibetan Plateau. The Cryosphere Discussion. doi.org/10.5194/tc-2022-178, 2023.

Monnier, S. and Kinnard, C.: Internal structure and composition of a rock glacier in the dry Andes, inferred from ground-penetrating radar data and its artefacts. Permafrost and periglacial processes, 26(4), 335-346, 2015.

Muller, S. W.: Permafrost, or Permanently Frozen Ground And Related Engineering Problems. Ann Arbour, J.W. Edward, 1947.

Müller, J., Vieli, A. and Gärtner-Roer, I.: Rock glaciers on the run – understanding rock glacier landform evolution and recent changes from numerical flow modeling. The Cryosphere 10, 2865-2886. doi.org/10.5194/tc-10-2865-2016, 2016.

---

## Author Comment (AC3)

*Response to reviews*

*We thank the reviewers for their constructive feedback, which helps optimizing our contribution and invited perspective. Our response is given below in italics. The italics in Review 1 were replaced by <> marks.*

**Review 1:**

[RG - rock glaciers, DG - debris covered glaciers]

**Summary**

This invited perspective manuscript discusses the application of criteria for identifying/detecting rock glaciers for two example sites, Gruben and Yerba Loca. A main focus is given to differentiating RG from DG. Following the introduction, Section 2 details how the criteria are applied to the Gruben site to distinguish the Gruben rock glacier from the neighbouring debris covered glacier. Section 3 discusses more complicated cases (contact zones of surface and subsurface ice) and concludes that <"neither the term "rock glacier" nor the term "debris-covered glacier" would be appropriate for such complex contact zones with their characteristically diffuse landforms.">

Section 4 critically states that contrasting views regarding the definition and genesis of rock glaciers exist in the scientific community, particularly regarding how RG differ and should be differentiated from DG. This section functions as a starting point for the following sections, which compare and contrast specific characteristics of RG and DG: Section 5 highlights that RG move as a result of viscous creep of permafrost. The "coherent movement pattern" of the creeping permafrost is transmitted to the surface debris, which is "interlocked" with the deeper layers. This results in characteristic furrows and ridges. In contrast, the surface debris of DG is not "interlocked" with the ice and there are no "coherent flow patterns" on the surface. Section 6 highlights that RG have oversteepened advancing fronts while advancing DG often show massive ice at the terminus. The Yerba Loca site is used as an example illustrating oversteepened fronts at RG and other features of creeping permafrost. Section 7 focuses on differences in the response of RG and DG to climate warming.

Section 8 reiterates key points made in the introduction and throughout the manuscript and concludes that RG and DG should and usually can be clearly distinguished based on the strategies developed by the IPA. Further, as stated in the introduction, the authors recommend that RG as permafrost phenomena remain under the purview of the GTN-P, while DG remain part of the GTN-G.

**General comments**

This is an important contribution to the ongoing discussion on identifying rock glaciers, e.g. for inventory purposes, and distinguishing them from debris covered glaciers and other landforms. A consistent approach and clear definitions are needed for RG inventorization and the RGIK/IPA guidelines provide a broadly applicable supporting framework for such efforts, many of which are currently ongoing in mountain regions around the world. The manuscript makes valuable points about separating RG from DG and will further support the development of community strategies and guidelines in this area. I am sure my comments/questions can be resolved and look forward to seeing this published in TC.

*Response: This perfectly summarizes our contribution. Thanks. We emphasize that advancing rock glaciers have talus-type fronts. This in fundamental contrast to the ice fronts of advancing debris-covered glaciers.*

Terminology and definitions:  In my opinion it would be beneficial to explicitly include the wording of the guidelines/criteria that are applied, tested and referred to throughout, as well as clear definitions of RG and other terms. Perhaps the introduction could be expanded by a short dictionary style section listing key terminology. This could also include the brief comment on subsurface and surface ice currently in the supplement.

*Response: We follow this well-taken suggestion by adding a new section about terms, characteristics and guidelines (subsequent sections are renumbered accordingly):*

*2. Terminology, characteristics and guidelines:*

[revised manuscript text omitted]

The authors point out repeatedly that DG remain DG no matter what, given that they do not turn into RG (a central message of this manuscript). Accordingly, small dead ice bodies that can remain when glaciers melt and may be partially or completely debris covered are still considered DG ("glaciers") in the sense that they are "not rock glaciers". However, they are also no longer "glaciers" in the typical sense. The authors sometimes use the more descriptive phrase "debris covered surface ice" or similar for such cases, but the usage is not always consistent. Defining "glacier" in some way or always using variations of "surface ice" in the context of "complex cases" could help prevent terminology related confusion. The authors also mention a size limit that separates glaciers from "not glaciers" in glacier inventories. It would be helpful to state what this limit is. The readership of TC can certainly make an educated guess about this and other matters, but the clarity of the manuscript could nonetheless be improved by adding some definitions and consistently using the

respective terminology.

*Response: Right – thanks. An explanation concerning various forms of surface ice and related minimum sizes as applicable in inventories is now provided in the new section 2.*

Complex cases, ambiguous landforms: The authors state in the introduction:

<An objective way of differentiating corresponding landforms and kinematics is essential in creating clarity when utilizing such landforms to assess where and how climate change impacts our planet, specifically the cryosphere, or when used in a regulatory/legal context, for example in view of hydrological significance; or generally, to avoid confusion and duplication.">

It is certainly important to distinguish RG from DG for these purposes. It is also important to have a practical and consistent way of dealing with ambiguous landforms that cannot easily be categorised as either DG or RG particularly for inventorization purposes. When inventorization of cryospheric features is connected to regulatory measures, the measures in question often do not themselves give clear definitions of what exactly should and should not be included in the inventory. Having a community consensus on landform definitions beyond RG and DG and on dealing with ambiguity would be beneficial in such cases. Do the authors have a recommendation or further comments on how to deal with ambiguous landforms (buried surface ice as well subsurface ice other than RG) from an inventory perspective, or how a consensus based community strategy to this end might be developed?

*Response: Concerning complex contact or transitional zones between surface ice and creeping perennially frozen ground and rock glaciers, RGIK rightly formulates that "the delimitation between the glacier or the ice patch section and the rock glacier section is not feasible without further direct or geophysical prospection". Such contact zones are not usually included in inventories of visible glaciers/surface ice and should not be part of rock-glacier inventories either. The latter are inventories of well-recognizable landforms, not of permafrost zones.*

I do not fully understand what the authors are suggesting related to the application of a size threshold and would appreciate more detail on this. Is the idea that a threshold (smaller than glaciers, thinner than underlying permafrost) helps identify the "complex cases" in a general sense, or should such thresholds be used as additional criteria alongside those of the RGIK/IPA for classification purposes? If the latter, how would this size limit be applied practically, for example when compiling an RG inventory? Should the excluded features be ignored in inventorization efforts?

*Response: Features which are not clearly identifiable as surface ice or as rock glaciers should indeed be excluded from inventories. This is common practice in glacier inventories and also recommended for rock-glacier inventories*

Contrasting views: The authors point out work by others that presents somewhat differing views on the distinction between RG and DG and RG genesis. Section 4 in particular is critical of these works. The authors have strong arguments that can stand alone and are not further strengthened by dismissive comments towards others. I would suggest revisiting this section and either expanding the overview of contrasting work for a more comprehensive picture (e.g. Knight et al, 2019; Jones et al, 2019, and others) or finding a more concise way of introducing the following sections.

*Response: Within the framework of policy-relevant global climate-system observation, scientific contributions and practical work must strictly be based on knowledge and understanding from measured facts. In this sense, the text was reformulated in order to avoid confusing discussions about "views" or "opinions", which remain unrelated to, and mostly even in full contradiction with quantitative data from field measurements (drilling, geophysical soundings) about subsurface thermal and ice conditions with their fundamental impact on material characteristics and related physical processes. Reference is, however, made to the literature overview by Janke and Bolch (2022) and to the community comments submitted by Harrison and Whalley/Azizi with response from our side. The references in our contribution concern publications, which report measured evidence.*

**Specific comments: Introduction:** For clarity and to help the reader, I suggest including:

    ● the specific "proposed technical recommendations/guidelines" that are to be tested. Since the stated aim of the manuscript is to test and comment on the guidelines, it would be useful to explicitly mention what these guidelines are.

    ● the "technical definition of rock glaciers'' used by the RGIK/IPA, assuming the authors agree with this definition. A reference to section 4 of the manuscript's supplement (surface vs subsurface ice) could be added alongside the definition of rock glaciers to set the stage for the surface/subsurface arguments that follow in the later sections. Alternatively, the short paragraph explaining this as currently contained in the supplement could simply be added to the main text.

*Response: Thanks. The new section 2 does this in a short and summarizing way.*

The two RGIK documents cited by the authors list two mandatory "geomorphological criteria'' for rock glacier detection (front and lateral margins) and one optional criterion (ridge-and-furrow topography, section 3a in the RGIK (2022) baseline concepts). The manuscript discusses RG fronts (Section 6) and ridge-furrow topography (Section 5) but does not mention the lateral margins criterion. If the aim is to test the RGIK criteria, a brief explanation of why lateral margins as a mandatory criterion are being excluded from this exercise of testing and commenting would be

helpful. If criteria other than those listed in the RGIK document are being tested, please clarify.

*Response: In the new section 2 we state that our focus is on frontal characteristics rather than lateral margins. As the Yerba Loca site documents, striking frontal characteristics (continued oversteepening and destabilisation, exposure of fresh debris) can also indicate active creep and advance of frozen debris outside clearly defined rock-glacier landforms. In the Gruben case, steep lateral margins mark the overall convex landform but the most striking indication of active creep movement is the advancing, oversteepened front.*

**Section 2: Rock glacier and cold debris-covered glacier at the Gruben site**

L95 <The differences are obvious, and the morphological criteria proposed by the IPA action group are adequate> Consider specifying what these criteria are.

*Response: Thanks. The formulation is now " ... the morphological criteria (steep front and lateral margins, surface pattern with ridges and furrows for rock glaciers, etc.) proposed by ...". More information is in the sequence which follows.*

Fig 2: I suggest adding a reference to the very helpful Fig. Sup.-1 in the supplement to the caption of this figure.

*Response: This is original work by one of the co-authors as part of the PhD thesis of JW.*

**Section 3: Complex zones with contacts between surface and subsurface ice**

L 125 <not every piece of surface ice is a "glacier"> It might be useful to briefly define your usage of the term "glacier" here or earlier in the manuscript, see general comment.

*Response: This is now done in the new section 2.*

L 139 <mostly smaller than the lower size limit applied to the term "glacier" in glacier inventories> For clarity, please state what this size limit is / give citations. Not all glacier inventories use consistent size limits and some make a case for including very small, debris covered ice bodies in regions where deglaciation is imminent. (See, e.g. the discussion in Section 5.4 of Fischer et al (2021))

*Response: The size limit is now indicated in the new section 2. The inclusion of even smaller bodies of surface ice in glacier inventories should not apply the term "glacier" but more appropriate terms like "perennial ice patches", "dead ice remains" or the like.*

**Section 4 - debris covered glaciers remain debris covered glaciers**

L 148: What would the authors consider "adequate" in this context? Perhaps a statement could be made on minimum required quantitative information.

*Response: This statement was eliminated.*

L151: can DG turn into "complex cases" of buried surface ice in contact with permafrost features as discussed in the previous section as a third option? If so, should that be added to the list as option C? If not it would be useful to briefly clarify.

*Response: Where debris-covered surface ice is in contact with permafrost, complex contact zones often develop. RGIK precisely describes such cases. The former contact zone between the polythermal Gruben glacier and the perennially frozen Gruben rock glacier is a well-documented example. The main source of confusion are "either-or" type discussions – "is  it a glacier or is it permafrost?" – which also constitutes the basis of the sometimes proposed equifinality concept of rock glacier origins. We now mention that this aspect is explicitly treated in the community comments with our response to them.*

L153 <lavastream-like>  Consider omitting in the interest of precise usage of terminology.

*Response: Accepted and omitted.*

L163 <Nevertheless, it is useful to understand the reason why this is the case ...>  Why what is the case? I find this sentence hard to follow, consider rephrasing. See also the general comment.

*Response: Reformulated:  " ... why version (b) does not seem to occur in nature and what ..."*

**Section 6 ... as visible at advancing fronts...**  L217: <Advancing debris-covered glaciers have become exceptional under conditions of atmospheric temperature rise and predominating glacier shrinkage. In such increasingly rare cases, massive ice of the flowing glacier is usually visible at near- vertical fronts where debris cannot accumulate (Figure 4)>  Does this imply that Belvedere Glacier (Fig 4) is currently advancing? Afaik that is not the case.

*Response: Ghiacciaio del Belvedere was intermittently advancing when the picture was taken from a helicopter. This is now mentioned in the figure caption.*

L220: <The advantages of adequately interpreting over-steepened fronts of creeping frozen talus/debris can be illustrated with an example from the Andes....>  Fig 5 shows oversteepened fronts marked by yellow arrows at rock glaciers (1, 2, 3) and ambiguous landforms (4, 5). What information is gained by "adequately interpreting" the oversteepened fronts and what does "adequate" mean in this context? Perhaps it would be informative to walk through all three of the RGIK/IPA

criteria for rock glacier detection (fronts, lateral margins, furrow-ridge topography) for the Yerba Loca landforms to clearly show how the criteria can help separate rock glaciers from other landforms.

*Response: "adequately" was eliminated. On line 226 we added after "rock glaciers" " … with steep fronts/lateral margins and recognizable ridge-and-furrow surface morphology …".*

**Section 7 … and its effects on ice loss as a response to long-term warming trends** L267: <…and hardly ever appears at advancing fronts> Are there studies showing massive ice at advancing fronts? citations?

*Response: We are not aware of any occurrences of large bodies of massive ice at undisturbed rock glacier fronts. Bodies of massive ice up to the meter range have been documented in a deep excavation at a rock glacier front (Fisch et al. 1978).*

**Section 8**

L277 <The rich available quantitative knowledge basis from borehole and geophysical data in combination with advanced material- /process-related understanding enables safe and straightforward discrimination between rock glaciers as viscous creep phenomena in ice-rich mountain permafrost and debris-covered glaciers. The corresponding strategies recommended by experts of the International Permafrost Association are clear and easy to follow.>

I suggest briefly stating again what the recommended strategies are. Borehole and geophysical data are available only for a small fraction of RG, DG, and other landforms. It could be pointed out that since the RGIK/IPA strategies are informed by and developed based on the process understanding and rich quantitative knowledge basis the authors refer to, they may be especially helpful in cases when inventories are being compiled without comprehensive geophysical information or boreholes.

*Response: Thanks for this suggestion which we gratefully take over. We now write " … corresponding strategies recommended by experts of the International Permafrost Association are informed by and developed based on the process understanding and rich quantitative knowledge basis from numerous sophisticated field investigations using advanced technologies. They are clear and easy to follow, and may be especially helpful in cases when inventories are being compiled without comprehensive site investigations including geophysical soundings or boreholes.*

L283 <Complex contact zones of surface ice (mostly perennial snow and ice patches, glacierets or small glaciers) with creep phenomena in ice-rich permafrost in cases constitute diffuse landforms beyond "either-or"- type landform classification.>

I understand that an in-depth discussion may be beyond the scope of this manuscript, but would the authors consider these ambiguous landforms the

responsibility of the GTN-P or GTN-G? How should they be inventoried? I agree that they are "beyond simplistic landform attribution" but they are relevant for inventories as assessments of the changing cryosphere in regulatory and or hydrological contexts.

*Response: It is our clear opinion that they should be excluded from glacier and permafrost inventories of exactly defined features but deserve more quantitative research. We explicitly state this in our response to the community comments and added in the summary and recommendations section of our contribution: "Exploring contacts and combinations of surface and subsurface ice with their strikingly different response characteristics concerning atmospheric warming is now indeed a growing field of advanced research. It involves quantitative treatment of the involved material properties and processes. This by far exceeds the possibilities of speculative interpretations based alone on visual surface inspection. A recent example illustrating the potential of multimethod field measurements to be used in such complex cases is the comprehensive investigation at the Chauvet site in the French Alps (Cusicanqui et al. 2023)."*

Personally, I would like to see the conclusion link back to the stated aim of the introduction, i.e., testing the application of the guidelines of the RGIK/IPA at the Gruben and Yerba Loca sites. Perhaps a brief summary pertaining to this aspect of the manuscript could be added, maybe with some generalised conclusions regarding the usefulness of the specific mandatory and optional criteria (as per RGIK 2022, 3a) that can be derived from the characteristics of the two test sites, i.e., RG and DG in close vicinity at Gruben and different kinds of creeping permafrost at Yerba Loca.

*Response: This suggestion is fine. We added the following paragraph at the beginning of the final section: "A combination of striking morphological and dynamic characteristics makes the difference between rock glaciers as viscous creep features in mountain permafrost and debris-covered glaciers (and smaller forms of surface ice) under conditions of ongoing global warming: convex versus concave shape, sharp versus diffuse edges, structured versus chaotic surfaces, continued coherent flow and advance versus slowing-down, disintegration and down-wasting. The test at Gruben and Yerba Loca illustrates the applicability of such criteria in concrete climate-related inventory and monitoring work but also indicates limits and complexities needing further exploration."*

**Typos and such:**

L161 & L271: I suggest replacing "safely" with "definitively" or a similar word. L261: Check spelling of Moelg/Mölg in citation

*Response: Thanks, done. "safely" was eliminated.*

**Supplement:**

Part 2, kinematics Yerba Loca: Can you indicate which of the numbered features (1-5) in Fig 5 are shown in Fig.-Sup. 1-3? Fig. Sup-2: typos in the caption, "Sub" instead of Sup

*Response: Thanks, done*

Part 4: not referenced in the manuscript? I suggest moving this forward.

*Response: Part 4 is now in the main text (new section 2).*

Honestly, when I agreed to give my opinion on this contribution, I feared that it would be yet another paper on the old discussion about rock glacier vs debris-covered glacier, which has been ongoing in the cryosphere scientific literature for several decades.

Instead, I found the paper very topical, because the authors' opinion is documented by data collected both decades ago but also in the last years with the employment of up to date techniques. Furthermore, the authors clearly state why it is still

important to clearly disentangle the two landforms. There is no doubt that the paper is written in a clear, concise manner and with a very logical structure.

I believe it is a paper that clearly exposes in an extremely effective way the authors' opinion about the distinction between rock glacier and debris-covered glacier. The completeness and clarity of the authors' statements make possible counter-deductions and different interpretations of the same or further data by those who have different opinions. And I believe that this type of paper also serves to stimulate a scientific discussion, free of misconceptions, simplifications, and genericity.

My modest experience in permafrost subject leads me to agree with the authors, although I have almost always had to deal with rock glaciers and little with debris-covered glaciers. I too, as a geomorphologist, believe that in many cases the landforms interpretation must go beyond mere intuitions supported by qualitative observations, even if sophisticated and reasonable, or non-decisive data, but that it is necessary to rely on measurements (or better sets of multi-method measurements) when the understanding of the formation mechanisms is complex and includes depositional and post-depositional (i.e. deformative) processes affecting mechanically thermally inhomogeneous materials.

I agree with the authors that the contact zone between debris-covered glacier and rock glacier is pivotal, both for a complete understanding of the differences among the two landforms, but also for dispelling doubts that may arise from the detection (instrumental or visual) of massive ice buried in the apical area of a rock glacier. In these regards, I would like to suggest the authors to clarify better how bodies of massive ice can be "transferred" from a (debris-covered) glacier to a rock glacier. Are they "syngenetically" incorporated by permafrost creep involving the marginal (ice-cored) deposits of a glacier? Is this the consequence of a glacier overlapping onto the root of a rock glacier? Can a fragment of ice core embedded in a rock glacier be displaced by permafrost creep also toward the mid-frontal parts of a rock glacier? These clarifications could explain how in various geophysical soundings values interpretable as massive ice have been identified in non-apical parts of rock glaciers, fuelling interpretations shifted towards debris-covered glaciers origin.

*Response: Thanks for these clear and encouraging words. We added the following paragraph in the section about complex cases: "Concerning the "transfer" of surface ice to creeping (rock glacier) permafrost, there is no simple or straightforward general solution. The Gruben and Yerba Loca examples, however, provide some indications. As mentioned in the caption of Figure Sup.-1, the isolated bodies with resistivities in the low MΩm range, still existing today on top of near 0°C permafrost in the former marginal zone of the LIA glacier are most probably dead ice from the small northern tributary underneath the Senggchuppa slope but could also be remains of a buried and frozen avalanche cone at the origin of the photogrammetrically defined flowlines. The earlier visible surface ice at Yerba Loca cannot be called "glaciers" for reasons of size but are/have been perennial ice patches, mostly from avalanche cones. In both cases,*

*Gruben as well as Yerba Loca, the buried ice bodies are more or less passively riding on top of thick perennially frozen sediments.*

About the effect of thermal protection acted by the active layer of rock glacier , I would suggest to complete the explanation by adding how the active layer can continue to grow if it is its thickening that makes the degradation of the permafrost increasingly slower.

*Response: We now added the following statement: "Further, a thickening of the coarse active layer has a substantial impact on the heat transfer between the atmosphere and the permafrost. First, the thermal resistance of the active layer increases as the thickness of the typically unsaturated debris layer increases. Air has a much lower thermal conductivity than ice or frozen ground (e.g., Andersland and Ladanyi, 2003; Arenson et al., 2021), which is why the thermal conductivity of the active layer tends to decrease as a result of permafrost degradation, contrasting the cover of a debris-covered glacier that cannot change its thermal resistance over time. Secondly and potentially more importantly, a thickening of the dry and coarse active layer allows increased air flow and with that additional cooling through air convection (Wicky and Hauck, 2020). The Rayleigh number, which describes the potential and the strength of natural convection in porous media (Kane et al., 2001; Nield and Bejan, 2017), is directly dependent on the thickness of the active layer. As shown in Figure 3, natural convection can increase, or start to form in the thickening active layer of degrading rock-glacier permafrost over time, but remains unchanged for a debris-covered glacier."*

I hope these opinions of mine can be helpful,

*Response: They are indeed. Thanks again.*

Best regards

***References related to our response and not contained yet in our contribution***

*Andersland, O.B. and Ladanyi, B.: Frozen Ground Engineering (2nd ed.). Wiley, 2003.*

*Baral, P. and Haq, M.A.: Spatial prediction of permafrost occurrence in Sikkim Himalayas using logistic regression, random forests, support vector machines and neural networks. Geomorphology 371, 107331. doi.org/10.1016/j.geomorph.2020.107331, 2020.*

*Bertone, A., Jones, N., Mair, V., Scotti, R., Strozzi, T. and Brardinoni, F.: A climate-driven, altitudinal transition in rock glacier dynamics detected through integration of geomorphological mapping and InSAR-based kinematics. The Cryosphere Discussion. doi.org/10.5194/tc-2023-143, 2023.*

*Cogley, J. G., R. Hock, Rasmussen, L. A., Arendt, A. A., Bauder, A., Braithwaite, R. J., Jansson, P., Kaser, G., Möller, M., Nicholson, L. and Zemp, M.: Glossary of Glacier Mass Balance and Related Terms, IHP-VII Technical Documents in Hydrology No. 86, IACS Contribution No. 2. P. UNESCO-IHP, Paris, 2011.*

Cusicanqui, D., Bodin, X., Duvillard, P.-A., Schoeneich, P., Revil, A., Assier, A., Berthet, J., Peyron, M., Roudnitska, S. and Rabatel, A.: Glacier, permafrost and thermokarst interactions in Alpine terrain: Insights from seven decades of reconstructed dynamics of the Chauvet glacial and periglacial system (Southern French Alps). Earth Surface Processes and Landforms 48 (13), 2595-2612. doi.org/10.1002/esp.5650, 2023.

de Pasquale, G., Valois, R., Schaffer, N. and MacDonell, S.: Contrasting geophysical signatures of a relict and an intact Andean rock glacier. The Cryosphere, 16(5), 1579-1596. doi.org/10.5194/tc-16-1579-2022, 2022.

Halla, C., Blöthe, J.H., Tapia Baldis, C., Trombotto Liaudat, D., Hilbich, C., Hauck, C. and Schrott, L.: Ice content and interannual water storage changes of an active rock glacier in the dry Andes of Argentina. The Cryosphere, 15(2), 1187-1213. doi.org/10.5194/tc-15-1187-2021, 2021.

Haq, M.A. and Baral, P. : Study of permafrost distribution in Sikkim Himalayas using Sentinel-2 satellite images and logistic regression modelling. Geomorphology, 333, 123-136. doi.org/10.1016/j.geomorph.2019.02.024, 2019.

Hauck, C. and Kneisel, C. (2008). Applied Geophysics in Periglacial Environments. Cambridge University Press. doi.org/10.1017/cbo9780511535628

Hausmann, H., Krainer, K., Brückl, E. and Mostler, W.: Internal structure and ice content of Reichenkar rock glacier (Stubai Alps, Austria) assessed by geophysical investigations. Permafrost and Periglacial Processes, 18(4), 351-367, 2007.

Herring, T., Lewkowicz, A.G., Hauck, C., Hilbich, C., Mollaret, C., Oldenborger, G.A., Uhlemann, S., Farzamian, M., Calmels, F. and Scandroglio, R.: Best practices for using electrical resistivity tomography to investigate permafrost. Permafrost and Periglacial Processes, 34 (4), 494-512. doi.org/10.1002/ppp.2207, 2023.

International Permafrost Association [IPA] (2023). What is Permafrost? https://www.permafrost.org/what-is-permafrost, accessed, August 2023.

Kane, D.L., Hinkel, K.M., Goering, D.J., Hinzman, L.D. and Outcalt, S.I.: Non-conductive heat transfer associated with frozen soils. Glob. Planet. Change 29, 275–292. doi: 10.1016/S0921-8181(01)00095-99, 2001.

Li, M., Yang, Y. and Peng, Z., Liu, G.: Assessment of rock glaciers and their water storage in Guokalariju, Tibetan Plateau. The Cryosphere Discussion. doi.org/10.5194/tc-2022-178, 2023.

Merz, K., Maurer, H., Buchli, T., Horstmeyer, H., Green, A. G. and Springman, S. M. : Evaluation of ground-based and helicopter ground-penetrating radar data acquired across an Alpine rock glacier. Permafrost and Periglacial Processes, 26(1), 13-27. doi: 10.1002/ppp.1836, 2015.

Nield, D.A. and Bejan, A.: Convection in Porous Media. Cham: Springer. 2017.

Pavoni, M., Sirch, F. and Boaga, J.: Electrical and electromagnetic geophysical prospecting for the monitoring of rock glaciers in the Dolomites, Northeast Italy. Sensors, 21(4), 1294. doi.org/10.3390/s21041294, 2021.

---

## Author Comment (AC5)

**Response to editor recommendations**

*We thank the editor, Tobias Bolch, for his feedback and recommendations. Our response is given below in italics.*

Dear Professor Haeberli, dear authors,

First of all, I like to thank you for the important contribution to TC with its open discussion. This is another excellent example of the value of the open discussion where not only reviewers but also the community can post comments.

*Thanks.*

I have now read the reviews and the provided public comments in detail. The reviewers are overall supportive, but ask for some clarifications. I'd like to specifically mention two comments of Rev#01:

She/he suggested to add some clearer definitions of the terminology which I think would be valuable.

*We agree and now introduce a new section 2 about terms, geophysical characteristics and RGIK guidelines for landform interpretation.*

It might also be a good idea to revisit the term Ice-Debris landform as suggested by S. Harrison in his comment and also *Ice-Debris Complex* as used in my 2019 paper (Bolch et al., 2019, ESPL, which is already cited in your perspective paper).

*These two terms may have their place in a very general "overview" sense. The Gruben and Yerba Loca sites, for instance, with their multiple ice- and debris-related phenomena may collectively be called ice-debris complexes. Climate-related inventory and monitoring work relating to permafrost and glaciers, however, must apply more differentiated, precisely defined concepts as illustrated in the RGIK/IPA recommendations and in our invited perspective. This is the reason why they are not contained in the RGIK guidelines and are not commonly applied in research about mountain permafrost and related creep phenomena.*

She/he also suggested to provide a more detailed overview of the existing contrasting views. This is in line with my initial review where I wrote "I recommend to include a more critical discussion and the related papers by other groups with slightly different views (e.g., but not only, Whalley, 2020, who also discussed the Gruben site, Knight et al., 2019 and/or other work by S. Harrisons group)".

*The full comment of review 1 reads as follows:*

*Contrasting views:  The authors point out work by others that presents somewhat differing views on the distinction between RG and DG and RG genesis. Section 4 in particular is critical of these works. The authors have strong arguments that can stand alone and are not further strengthened by dismissive comments towards others. I would suggest revisiting this section and either expanding the overview of contrasting work for a more comprehensive picture (e.g. Knight et al, 2019; Jones et al, 2019, and others) or finding a more concise way of introducing the following sections.*

*We welcome this differentiated feedback, prefer to follow the second recommendation in the review, and therefore eliminate potentially dismissive statements with related references. Reference is now made to the literature overview by Janke and Bolch (which needs no repetition) and to the Harrison/Whalley community comments with their references, together with our response. With this, interested readers have access to the literature about contrasting views.*

Providing more detailed information and a more in-depth discussion about the contrasting views view would also be very beneficial regarding the community comments. The contribution by Stephan Harrison in general supportive and well written, while the comment by Brian Whalley is more critical. It is a perspective paper where it is fine to keep your opinion and I do not expect to consider all suggested references, but it would make the paper much stronger if the contrasting views would be better

presented and discussed in more depth. This would then help that the opinions "converge" and will in particular be helpful for those scientists that are now starting to investigate rock glaciers.

*The critically reflected test presented in our invited perspective concerns technical recommendations prepared by experts on behalf of international organizations and policy-relevant, climate-oriented programs. The focus is thereby on current/ongoing evolution of permafrost and related creep phenomena in cold mountains. The argumentation in our perspective as well as in the work of RGIK/IPA strictly relates to the available, quantitatively measured evidence concerning physical (especially thermal) conditions, subsurface ice characteristics, related material properties, flow processes, response to climate change, and the involved scales in space and time. There are no contrasting "opinions" or "views" about the related measurement-based findings. Rather than finding a compromise between diverging theoretical opinions about "landform origins", our aim and the strategy of RGIK/IPA is to build on the scientific consensus about the rich existing measured evidence.*

I am inviting you to provide a point-to-point reply to all detailed comments by the reviewers and also a detailed reply to the community comments. I will then make a decision how to proceed.

Thank you again for choosing TC for your perspective and best regards,

Tobias Bolch - Editor

---

## Author Response (AR2)

***Author response (in italics) to:***

Editor decision: Publish subject to technical corrections
by [Tobias Bolch]
**Public justification (visible to the public if the article is accepted and published)**:
Dear Professor Haeberli, dear authors,

Thank you very much for carefully revision of the paper and addressing the comments leading to a clearly improved manuscript. I was asking for a 2nd review, but did not receive one yet. In order not to delay the process and in view of the favourable comments of the reviewers I decided to make a decision based on the available review.

*Thanks. This is appreciated.*

In line with the reviewer and my initial comments I like thank you for acknowledging the competing views, but also ask you to officially cite the community comments with a doi (at least the one which is written in a more objective way). In this regard I also ask you not to remove the reference to Whalley (2020); Geogr. Annnaler A. who provides a different interpretation of Gruben rock glacier.

*The CCs are now cited with the related doi in the reference list. The reference to Whalley (2020) is re-introduced together with a reference to Anderson et al. (2018) postulatimng similar beliefs.*

Few further specific minor comments. (Rem.: The line number are taken from the track changes manuscript.

L. 49: I suggest one more recent paper from the observational side, e.g.
Neckel, N., Loibl, D., Rankl, M., 2017. Recent slowdown and thinning of debris-covered glaciers in south-eastern Tibet. Earth and Planetary Science Letters 464, 95–102. .

*Thanks. Neckel et al. (2017) is now cited and added in the reference list.*

L. 53: Here and later: You may refer to the most recent version which has just been released: https://doi.org/10.51363/unifr.srr.2023.002

*All references to RGIK are adjusted accordingly*

L 76 ff: Definition of rock glaciers. Refer to two or three references to back up your definition.

*Haeberli et al. (2006) and Berthling (2011) have been added. Haeberli et al. (2006) is a task force report of ICSI (now IACS) and IAP.*

L: 186: Should be Bolch et al. (2019). This is the official year of publication. Please also adjust the reference list accordingly.

*This is adjusted in the text and in the reference list.*

Data availability section: The current statement is too vague. You need to provide more detailed information where and when the data will be available. See https://www.the-cryosphere.net/policies/data_policy.html

*More detailed information was added. The resistivity data are available from Julie Wee. Publication in the Cryosphere is in preparation.*

Author contribution: I recommend to specifically mention the contribution of all authors. Acknowledgements. Thank you for mentioning the editor. There is no need for it as the scientific editor is anyway mentioned beneath the Acknowledgement section.

*The specific role of Nico Mölg is now explicitly defined and mentioning of the editor is eliminated in the acknowledgements.*

Please also address the further minor comment of the reviewer.

*All reviewer comments had already been addressed.*

I selected technical corrections as I trust that you will address all the remaining minor comments.

*Thanks.*

Thanks again for choosing TC for your perspective making it a nice example of the value of the open discussion.

*We agree and highly appreciate the open review system of The Cryosphere.*

Best regards,
Tobias Bolch